# COMPUTERRL: SCALING END-TO-END ONLINE REINFORCEMENT LEARNING FOR COMPUTER USE AGENTS

**Hanyu Lai**[1*†], **Xiao Liu**[1,2*], **Yanxiao Zhao**[3*†], **Han Xu**[2], **Hanchen Zhang**[1†], **Bohao Jing**[2†],
**Yanyu Ren**[1†], **Shuntian Yao**[1†], **Yuxiao Dong**[1], **Jie Tang**[1]

[1] Tsinghua University   [2] Z.AI   [3] University of Chinese Academy of Sciences

## ABSTRACT

We introduce COMPUTERRL, a framework for autonomous desktop intelligence that enables agents to operate complex digital workspaces skillfully. COMPUTERRL features the API-GUI paradigm, which unifies programmatic API calls and direct GUI interaction to address the inherent mismatch between machine agents and human-centric desktop environments. Scaling end-to-end RL training is crucial for improvement and generalization across diverse desktop tasks; however, it remains challenging due to environmental inefficiency and instability during extended training. To support scalable and robust training, we develop a distributed RL infrastructure capable of orchestrating thousands of parallel virtual desktop environments to accelerate large-scale online RL. Furthermore, we propose Entropulse, a training strategy that alternates reinforcement learning with supervised fine-tuning, effectively mitigating entropy collapse during extended training runs. We employ COMPUTERRL on open models GLM-4-9B-0414 and GLM-4.1V-9B-Thinking, and evaluate them on the OSWorld benchmark. The GLM-COMPUTERRL-9B achieves a new state-of-the-art accuracy of **48.9%**, demonstrating significant improvements for general agents in desktop automation. Our code is available at https://github.com/THUDM/ComputerRL.

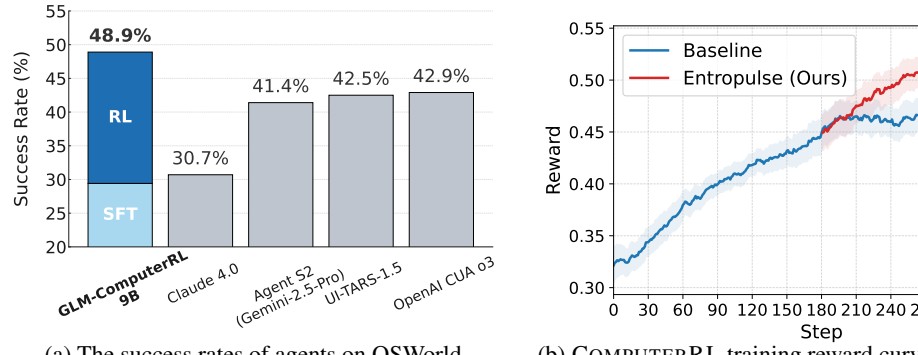

(a) The success rates of agents on OSWorld.   (b) COMPUTERRL training reward curves (95% CIs).

Figure 1: COMPUTERRL enables efficient end-to-end online policy optimization for OS agents. (a) On OSWorld (Xie et al., 2024), GLM-COMPUTERRL, trained with COMPUTERRL, outperforms state-of-the-art agents. (b) Our Entropulse approach yields higher average training rewards and improves both learning efficiency and final performance over conventional methods.

## 1 INTRODUCTION

Large Language Models (LLMs) (Achiam et al., 2023; Touvron et al., 2023b; Zeng et al., 2022; GLM et al., 2024; Team et al., 2023; Guo et al., 2025; Bai et al., 2023a; Yang et al., 2025) have dramatically

---

*HL, XL and YZ contributed equally.
†Work partially done while these authors interned at Z.AI. Corresponding author: JT

expanded the scope and depth of artificial intelligence capabilities, driving a profound re-examination of our understanding of machine intelligence. Among all scenarios, the emergence of LLM-based GUI (graphical user interface) agents, capable of independently perceiving, reasoning, and executing complex tasks on user devices, has aroused particular interest from researchers (Xi et al., 2023; Wang et al., 2023; Liu et al., 2023). Given that desktops remain central to intelligence-intensive tasks, developing computer use agents is crucial for fundamentally transforming human-computer interactions and elevating AI capabilities (Agashe et al., 2025; Wu et al., 2024a).

Despite previous attempts to develop computer use agents (Agashe et al., 2025; Lei et al., 2024; Xie et al., 2025), enabling them to operate autonomously over extended periods in real-world scenarios remains a significant challenge. The first primary obstacle arises from the fact that GUIs are inherently designed for human interaction, making the simulation of human actions by GUI agents (Liu et al., 2024b; OpenAI, 2025; Qin et al., 2025) a non-trivial and cumbersome endeavor. Second, current mainstream approaches of behavior cloning (BC) (Bain & Sammut, 1995; Bratko et al., 1995), including manual annotation (He et al., 2025) and model distillation (Sun et al., 2024; Xu et al., 2024), are limited in scalability and effectiveness. Manual annotation, while precise, is prohibitively labor-intensive for complex tasks. Model distillation, on the other hand, is constrained by the performance of the teacher models, limiting overall capability. Both methods typically exhibit poor generalization and limited error recovery abilities. Finally, although reinforcement learning (RL) has shown potential for desktop automation tasks (Lu et al., 2025; Feng et al., 2025), its practical application remains restricted due to computational complexity and methodological challenges. Complex environments, slow convergence, and known inefficiencies in RL training (Xie et al., 2024; Bonatti et al., 2024; Yu et al., 2025; Fu et al., 2025) severely limit its large-scale adoption in training computer use agents.

In this work, we propose COMPUTERRL, an end-to-end algorithmic framework designed to advance desktop-level planning, reasoning, and device operation. This framework includes a new API-GUI interaction paradigm, a scalable RL training infrastructure for computer environments, and an RL algorithm for extended effective training. First, we introduce API-GUI, a large-scale, automatically constructed API ecosystem that enables the agent to transcend the inherent biases of human-oriented operational paradigms. It instead leverages a more machine-oriented approach for device interaction, which combines API calls and GUI actions, thereby significantly enhancing both the versatility and overall performance of the agent. Second, we develop a distributed training infrastructure utilizing virtual machine clusters based on Docker and gRPC protocols for scalability, which is fully compatible with AgentBench (Liu et al., 2023). This infrastructure supports **thousands of parallel environments**, ensuring high scalability and consistent interactions across all environments. Additionally, we integrate the training infrastructure with the AgentRL framework (Zhang et al., 2025) to facilitate efficient asynchronous training, thereby accelerating the training process. Finally, to counteract stagnation and convergence issues in RL training—specifically, entropy collapse and rising KL divergence—we propose Entropulse, which alternates between RL and SFT phases periodically. This approach maintains exploratory capacity and ensures continuous performance gains (Figure 1b).

As a result, by harnessing end-to-end RL and optimization in the desktop environment, COMPUTERRL has achieved remarkable improvement in understanding and operating GUIs. Evaluation on the OSWorld benchmark (Xie et al., 2024) shows COMPUTERRL's significant improvements (see Figure 1a) in computer use challenges, achieving a success rate of **48.9%** (with 66% performance gain from RL), outperforming other state-of-the-art models including OpenAI CUA o3 (42.9%), UI-TARS-1.5 (42.5%), and Anthropic Claude Sonnet 4 (30.7%).

In summary, our contributions are as follows:

- We propose a new interaction paradigm, a shift from human-centric to machine-oriented interaction by introducing a large-scale, automatically constructed API ecosystem integrated with conventional GUI operations. This approach addresses the inherent mismatch between human-designed interfaces and artificial agent capabilities, while achieving superior operational efficiency and generalization performance on computer-based tasks.
- We establish a large-scale, distributed RL infrastructure for computer use agents by reconstructing virtual machine clusters, achieving unprecedented scalability with thousands of parallel environments and seamless AgentBench compatibility, thereby overcoming the critical bottleneck that has limited RL-based computer use agent training to scale experiments and enabling breakthrough results in large-scale agent training.

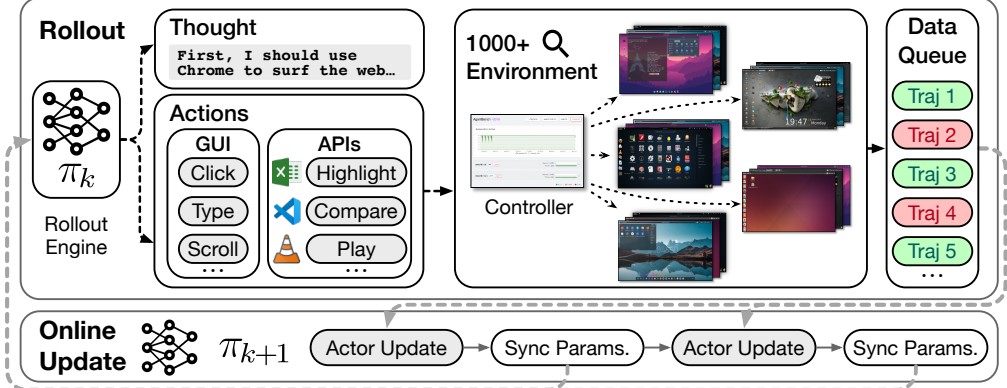

Figure 2: Overview of COMPUTERRL framework. We introduce an API-GUI action paradigm that seamlessly integrates automatically constructed APIs with GUI actions to improve agent efficiency and effectiveness. A large-scale parallel desktop environment with 1,000+ real-world instances, combined with an asynchronous RL framework, enables efficient sampling and robust agent training.

- We introduce Entropulse, a novel training methodology that systematically addresses the challenges of entropy collapse and KL divergence accumulation in extended RL training through strategic alternation between RL and SFT phases, enabling sustained performance improvements and achieving state-of-the-art performance in computer automation.

## 2 THE COMPUTERRL FRAMEWORK

The human-oriented design of GUIs hinders agent efficiency, while limited environment scalability restricts large-scale training. This section presents the COMPUTERRL framework (see Figure 2), which features an API-GUI paradigm that integrates human-like GUI interactions with efficient API invocation. Additionally, we develop a scalable Ubuntu desktop environment for parallelism and utilize a fully asynchronous RL framework for efficient training.

### 2.1 GENERAL API-GUI PARADIGM

Existing GUI agents face challenges due to their reliance on human-like interactions, while API-based control offers efficiency but introduces implementation complexity and security restrictions. To address these issues, we propose an API-GUI paradigm that unifies both action spaces, enabling agents to leverage API efficiency while retaining GUI versatility.

We develop an LLM-based automated workflow for application API development (Yang et al., 2024a; Wang et al., 2024), significantly lowering the barrier for API creation. Users provide exemplar tasks, and our system autonomously generates API code and test cases through three stages:

- **Requirement Analysis**: Users provide task examples for the target application. Our LLM analyzes these instances, extracts essential functionalities, and compares against existing API interfaces to identify gaps. New interfaces are automatically generated for uncovered functionalities, with a focus on general-purpose functions to minimize complexity and enhance usability.
- **API Implementation**: The workflow iterates over each interface definition, implementing API functionalities using designated Python libraries. Error-handling mechanisms and logging are implemented for debugging and maintenance purposes.
- **Test Case Generation**: Similar to Li & Yuan (2024), we verify API correctness by checking: (1) runtime error-free invocation and (2) correct results across parameter inputs; failed APIs receive error feedback for autonomous correction.

This methodology enables the creation of application-specific APIs with minimal human intervention. We have developed API sets for multiple Ubuntu applications and validated their effectiveness through experiments. Detailed API development workflow is provided in Appendix A. The agent action space and prompt formulation are detailed in Appendix B and C.

## 2.2 STABLE UBUNTU ENVIRONMENT FOR LARGE-SCALE PARALLELISM

A stable and scalable Ubuntu environment is essential for constructing behavior cloning data and large-scale RL training. Building on OSWorld (Xie et al., 2024), we identify key limitations:

- **Resource Intensiveness and Stability**: VMs are CPU-intensive and unstable under high concurrency, causing performance degradation and system freezes.
- **Network Bottlenecks**: Heavy workloads cause network overhead, connection failures, and IP address loss, hindering agent interaction and logging.
- **Lack of Native Distributed Support**: OSWorld lacks multi-node clustering support, preventing efficient distributed deployment.

To address these limitations, we build a robust and parallelizable OSWorld infrastructure (see Figure 2) with the following innovations:

- **Standardized, Decoupled Interface**: We refactor the environment via AgentBench API, providing a unified interface that decouples environment execution from the computational back-end and enables flexible resource management.
- **Lightweight VM Deployment**: Using `qemu-in-dockere`, we deploy containerized Ubuntu VMs with streamlined images that reduce network issues and optimize resource usage, significantly lowering per-instance CPU consumption.
- **Distributed Multi-Node Clustering**: We employ gRPC-based communication to link CPU nodes into a distributed cluster with centralized resource allocation and orchestration.
- **Web-based Visualization and Monitoring**: A web interface provides real-time visualization of environment statuses, agent states, and resource allocation, improving usability and debugging capabilities.

Through these technical improvements, our system supports deployment of **several thousands** of concurrent environments on a multi-node CPU cluster, as validated by extensive empirical evaluation. Results confirm our platform's superior stability, resource efficiency, and scalability, making it an enabling infrastructure for large-scale RL and agent-based research.

## 2.3 FULL-ASYNCHRONOUS RL FRAMEWORK FOR EFFICIENT TRAINING

Existing RL frameworks rely on synchronous training paradigms, where rollout collection and parameter updates are alternated, resulting in training inefficiencies. To address the limitation, we use the AgentRL framework (Zhang et al., 2025) for fully asynchronous RL training with the following designs:

- **Resource Partitioning**: Data collection runs on dedicated resources while the trainer streams data from the replay engine, preventing mutual blocking.
- **Dynamic Batch Sizing**: The trainer processes incoming data with flexible batch sizes, reducing idle time and improving efficiency.
- **Modular Component Isolation**: Actor, reference, and critic modules run independently with dedicated resources. We utilize PyTorch distributed groups and NCCL for efficient parameter sharing.
- **Off-policy Bias Mitigation**: We limit the replay buffer size and sync trajectories after each update, ensuring trajectories remain close to the latest policy.

Through a stable, high-concurrency desktop environment and the decoupling of training from rollout, we markedly enhance the efficiency of sampling and RL training. Our system achieves a high average power consumption per GPU, reflecting optimal resource utilization. This design supports scalable, high-throughput RL training by enabling dynamic workload balancing, resulting in a significant improvement in hardware efficiency and overall training throughput.

## 3 THE COMPUTERRL TRAINING

In Section 2, we establish a robust foundation for large-scale agent training. However, scaling end-to-end training still faces challenges in initializing a capable base policy and entropy collapse

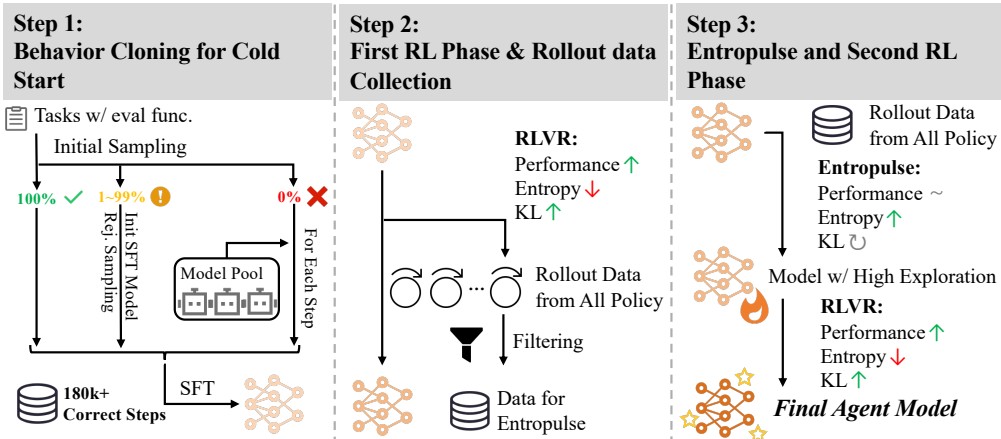

Figure 3: Overview of COMPUTERRL, which includes three stages: (1) BC cold start with trajectories collected from general LLMs; (2) RL with step-level GRPO using verifiable, rule-based rewards; (3) Entropulse, which alternates RL with SFT on correct rollouts to restore entropy and sustain learning.

during RL. This section details our scalable COMPUTERRL training approach and its algorithmic innovations for extended training in desktop environments.

## 3.1 BEHAVIOR CLONING SETUP

To perform a cold start for our model, we employ BC as the initial stage of training. By imitating user interactions, BC enables agents to acquire foundational competencies, thereby facilitating rapid adaptation to computer operations and tasks.

**Trajectory Collection with Multiple LLMs.** We manually collect extensive tasks with corresponding evaluation functions (see Appendix F) and augment to construct an 8k-task dataset. However, the large-scale collection of high-quality trajectories remains challenging. Manual annotation is prohibitively expensive, and relying on a single model for trajectory generation results in limited and homogeneous data distribution constrained by that model's capabilities. To address these limitations, we leverage the complementary strengths of multiple advanced models to collect a diverse and high-quality set of interaction trajectories. Concretely, our data pipeline consists of three key stages:

1. **Initial Sampling**: For each task, we utilize closed-source LLMs to sample several trajectories per task independently. We record both the complete interaction trajectories and the outputs produced by the respective evaluation functions. This procedure yields a rich set of diverse trajectories that serve as the foundation for subsequent data augmentation and model adaptation.

2. **Outcome Stratification**: Following initial data collection, we perform a stratified analysis of task outcomes by categorizing all tasks into three groups based on achieved accuracy: **Fully Solved** ($\text{acc} = 100\%$), **Partially Solved** ($0 < \text{acc} < 100\%$), and **Unsolved** ($\text{acc} = 0\%$).

3. **Task-Oriented Augmentation with Stratified Sampling**: For partially solved tasks, we conduct SFT on our backbone model using the initial trajectories as input. The fine-tuned model is then used to sample additional trajectories for each task, thereby substantially expanding the coverage and quality of trajectories for tasks where model proficiency was previously limited.

   For tasks classified as unsolved, we build a model pool of high-performing models and randomly select one to determine each action. This approach leverages inter-model variance at the task level, as different models exhibit distinct areas of expertise despite comparable aggregate performance, enabling trajectory generation that is unattainable by any single model.

We systematically aggregate and filter the collected interaction data, retaining only successful trajectories (**180k+** correct steps), and employ them for supervised fine-tuning of the model. This strategy equips the model with robust desktop manipulation capabilities and foundational reasoning abilities, significantly enhancing the performance of the base model.

## 3.2 REINFORCEMENT LEARNING WITH VERIFIABLE REWARDS

**Step-Level Group Relative Policy Optimization.** We extend the GRPO algorithm (Shao et al., 2024) to the step-level, making it more suitable for agent RL training. For each task $\tau$, the policy $\pi_\theta$ interacts with the desktop environment and samples $G$ trajectories $\mathcal{T}_1, \mathcal{T}_2, \ldots, \mathcal{T}_G$. The $i$-th trajectory consists of $L_i$ step-level actions $o_{i,1}, \ldots, o_{i,L_i}$. All steps from the same task are grouped, and the advantage $A_{i,j}$ is computed for each step. The overall loss aggregates all step advantages as follows:

$$\mathcal{J}_{StepGRPO}(\theta) = \mathbb{E}_{\mathcal{T} \sim P(\mathcal{T}), \left\{ \{o_{i,j}\}_{j=1}^{L_i} \right\}_{i=1}^{G} \sim \pi_{\theta_{old}}} \left[ \frac{1}{\sum_{i=1}^{G} L_i} \sum_{i=1}^{G} \sum_{j=1}^{L_i} \left( \min \left( \frac{\pi_\theta(o_{i,j}|q_{i,j})}{\pi_{\theta_{old}}(o_{i,j}|q_{i,j})} A_{i,j}, \right. \right. \right.$$
$$\left. \left. \left. \text{clip}\left( \frac{\pi_\theta(o_{i,j}|q_{i,j})}{\pi_{\theta_{old}}(o_{i,j}|q_{i,j})}, 1 - \epsilon, 1 + \epsilon \right) A_{i,j} \right) - \beta \mathbb{D}_{KL}(\pi_\theta \| \pi_{ref}) \right) \right],$$
$$A_{i,j} = \frac{r_{i,j} - \text{mean}(\mathcal{R})}{\text{std}(\mathcal{R})}, \quad \mathcal{R} = \{r_{u,v} \mid u = 1, \ldots, G, \, v = 1, \ldots, L_u\}$$

**Reward Design.** We select a subset of the constructed human-annotated data (in Section 3.1) for RL and employ a rule-based verification function to provide verifiable training signals for each trajectory. Successfully solved trajectories receive a reward of 1 for every correctly formatted action that contributes to the solution; failed trajectories or improperly formatted actions receive a reward of 0. Unlike conventional approaches that propagate step-wise returns via the Bellman equation, our methodology treats each prompt-response pair as an independent training instance with rewards based on the final trajectory outcome. This direct reward assignment provides explicit feedback by coupling agent behaviors with task success, facilitating effective policy optimization.

## 3.3 ENTROPULSE FOR SCALING RL TRAINING

In the RL training in Section 3.2, we observe that model performance plateaus after hundreds of training steps, with stagnating task completion rates and decreasing entropy. This premature convergence motivates us to investigate strategies for extending effective training and enhancing policy exploration. Inspired by DAPO (Yu et al., 2025), we experiment with increasing the clipping threshold, which attenuates the decline in entropy but significantly slows down policy improvement.

To address the issue, we propose Entropulse, motivated by the observation that SFT and RL objectives differ markedly during training. As entropy decreases during RL optimization, integrating SFT at critical junctures enhances exploration and trajectory diversity, facilitating further policy optimization. During initial RL training, we aggregate and retain all successful rollout trajectories. While conventionally discarded after single use, these trajectories from various policies at different training steps represent valuable and diverse behavioral data.

We process this dataset by randomly selecting successful trajectories per unique task to construct a new SFT training set, which exhibits the following attributes:

1. **High quality**: All data comprises completed, high-fidelity trajectories.
2. **Diversity**: Rollouts originate from heterogeneous policies in different training steps, offering a variety of problem-solving strategies.
3. **Computational efficiency**: The dataset leverages existing interaction data, eliminating the need for additional environment rollouts.

SFT on this dataset produces notable shifts in policy behavior. While evaluation task performance remains stable, the resulting policy shows increased entropy relative to the original one, indicating enhanced exploration. Building upon this enhanced exploration capability, we conduct a second round of RL training, which yields significant performance improvements and enables us to achieve state-of-the-art results in computer automation. The training and hardware details are in Appendix D.

## 4 EXPERIMENTS

We employ COMPUTERRL on GLM-4-9B-0414 (GLM et al., 2024) and GLM-4.1V-9B-Thinking (Hong et al., 2025), to produce GLM-COMPUTERRL-9b. We conduct extensive ex-

Table 1: GLM-COMPUTERRL performance on OSWorld and OSWorld-Verified (updated in 2025.08). We compare GLM-COMPUTERRL with state-of-the-art agents, including both proprietary and open models.

| Agent Model | #Params | OSWorld | OSWorld-Verified |
|---|---|---|---|
| *Proprietary Models* | | | |
| Aria-UI w/ GPT-4o (Yang et al., 2024b) | - | 15.2 | - |
| Aguvis-72B w/ GPT-4o (Xu et al., 2024) | - | 17.0 | - |
| Claude 3.7 Sonnet (Anthropic, 2023) | - | 28.0 | 35.8 |
| Claude 4.0 Sonnet (Anthropic, 2023) | - | 30.7 | 43.9 |
| Agent S2 w/ Claude-3.7-Sonnet (Agashe et al., 2025) | - | 34.5 | - |
| InfantAgent (Lei et al., 2024) | - | 35.3 | - |
| OpenAI CUA 4o (OpenAI, 2025) | - | 38.1 | 31.3 |
| Agent S2 w/ Gemini-2.5-Pro (Agashe et al., 2025) | - | 41.4 | 45.8 |
| UI-TARS-1.5 (Qin et al., 2025) | - | 42.5 | - |
| OpenAI CUA o3 (OpenAI, 2025) | - | 42.9 | - |
| *Open Models* | | | |
| Qwen2.5-vl-72B (Bai et al., 2023b) | 72B | 8.8 | 5.0 |
| PC Agent-E (He et al., 2025) | 72B | 14.9 | - |
| UI-TARS-72B-SFT (Qin et al., 2025) | 72B | 18.8 | - |
| UI-TARS-72B-DPO (Qin et al., 2025) | 72B | 24.6 | 27.1 |
| UI-TARS-1.5-7B (Qin et al., 2025) | 7B | 26.9 | 27.4 |
| Jedi-7B w/ GPT-4o (Xie et al., 2025) | 7B+ | 27.0 | 29.3 |
| UI-TARS-7B-1.5 + ARPO (Lu et al., 2025) | 7B | 29.9 | - |
| COMPUTERRL *(ours)* | | | |
| w/ GLM-4-9B-0414 | 9B | 48.1±1.0 | 47.3 |
| w/ GLM-4.1V-9B-Thinking | 9B | **48.9±0.5** | **48.0** |

periments across various scenarios to evaluate GLM-COMPUTERRL's performance within the computer environment.

## 4.1 MAIN RESULTS

To closely reflect the real user experience, we evaluate GLM-COMPUTERRL on the OSWorld (Xie et al., 2024) and OSWorld-Verified benchmark, comparing its performance against state-of-the-art models, including CUA (OpenAI, 2025), Claude-4 (Anthropic, 2023), and UI-TARS (Qin et al., 2025), among others. The comparative results are in Table 1. The results indicate that GLM-COMPUTERRL achieves superior performance across a range of domains, with its advantages most pronounced in the challenging multi-apps setting. Moreover, by employing the API-GUI strategy, GLM-COMPUTERRL can accomplish tasks using at most **1/3** of the steps required by the strongest baseline approaches, demonstrating remarkable gains in execution efficiency. These results underscore the potential of COMPUTERRL to advance the state of the art in computer automation across various applications.

## 4.2 OFFICE APPLICATION PERFORMANCE

As a critical interface for delivering and presenting, office application constitutes an important testbed for evaluating computer use agents. To assess agent performance in this domain, we curate a set of 180 challenging tasks from three sources: SpreadsheetBench (Ma et al., 2024), PPTC (Guo et al., 2023), and in-house developed Writer domain tasks. These tasks are adapted as necessary to integrate them into the OSWorld framework. The resulting benchmark, termed **OfficeWorld**, enables systematic measurement of agent capabilities in office-oriented scenarios. The results are in Table 2.

## 4.3 ABLATION STUDY

To evaluate the influence of various algorithms and training datasets on agent performance, we present an ablation study on the OSWorld benchmark in Table 3.

**Framework Ablation.** We compare the performance of the GUI-only approach with our proposed API-GUI paradigm using GPT-4o. The results demonstrate that the API-GUI paradigm substantially outperforms the GUI-only baseline across all domains. Specifically, the API-GUI strategy achieves

Table 2: GLM-COMPUTERRL performance on OfficeWorld compared to common baselines. We employ the same framework (with tools) and test settings to ensure a fair comparison.

| Agent Model | Word | Excel | PPT | Average |
|---|---|---|---|---|
| DeepSeek-V3.1 (Liu et al., 2024a) | 6.7 | 35.0 | 21.7 | 21.1 |
| DeepSeek-R1 Guo et al. (2025) | 13.3 | 36.7 | 18.3 | 22.8 |
| Claude 3.7 Sonnet (Anthropic, 2023) | 15.0 | 25.0 | 25.0 | 21.7 |
| Claude 4.0 Sonnet (Anthropic, 2023) | 18.3 | 35.0 | 20.0 | 24.4 |
| Gemini-2.5-Pro (Team et al., 2023) | 5.0 | 11.7 | 20.0 | 12.2 |
| GPT-4o (Hurst et al., 2024) | 18.3 | 21.7 | 8.3 | 16.1 |
| GPT-4.1 (Achiam et al., 2023) | 21.7 | 25.0 | 28.3 | 25.0 |
| OpenAI o3 (Jaech et al., 2024) | 23.3 | 36.7 | 41.7 | 33.9 |
| COMPUTERRL (*ours*) | | | | |
| w/ GLM-4-9B-0414 | 21.7 | **58.3** | **43.3** | 41.1 |
| w/ GLM-4.1V-9B-Thinking | **30.0** | **58.3** | 41.7 | **43.3** |

Table 3: Ablation study on framework designs and training methods. We categorize OSWorld into five distinct domains to facilitate a granular comparison of different strategies across various domains.

| Method | OS | Office | Daily | Professional | Workflow | Avg. |
|---|---|---|---|---|---|---|
| Framework Ablation (w/ GPT-4o) | | | | | | |
| GUI Only | 41.7 | 6.2 | 12.3 | 14.3 | 7.5 | 11.2 |
| API-GUI | 52.6 | 27.9 | 25.7 | 41.6 | 10.8 | 26.2 |
| Training Ablation (w/ Qwen2.5-14B) | | | | | | |
| Untrained | 20.8 | 17.2 | 19.7 | 22.9 | 3.3 | 15.2 |
| +) Behavior Cloning | 54.2 | 35.0 | 37.2 | 45.8 | 10.8 | 31.9 |
| +) RL Phase 1 | 83.3 | 46.1 | 45.1 | 56.3 | 16.1 | 42.0 |
| +) Entropulse | 75.0 | 42.3 | 50.6 | 52.1 | 18.9 | 41.5 |
| +) RL Phase 2 | **83.3** | **46.2** | **46.7** | **60.4** | **27.2** | **45.8** |

an average success rate of 26.2%, representing a 134% improvement over the GUI-only approach (11.2%). The most significant gains are observed in the Office (27.9% vs. 6.2%) and Professional (41.6% vs. 14.3%) domains, where API-GUI provides 350% and 191% improvements, respectively. These results validate our core hypothesis that combining API calls with GUI interactions enables more efficient and reliable task execution, particularly for complex professional workflows that benefit from programmatic control.

**Training Ablation.** We study the progressive impact of different training stages using Qwen2.5-14B. Starting from the backbone, Behavior Cloning (BC) establishes a solid foundation with 31.9%. The first RL phase (RL1) yields substantial gains, increasing the performance to 42.0% (+10.1%). Interestingly, Entropulse phase maintains similar performance (41.5%) while significantly increasing action entropy, which enhances exploration diversity and enables the final RL2 phase to achieve further improvements. The RL2 phase achieves the best performance at 45.8% (+3.8% from RL1), benefiting from the increased exploration capacity introduced by Entropulse. Notably, the Workflow domain shows the most dramatic improvement throughout training (10.8% → 27.2%), while the other domains maintain consistently high performance, highlighting the importance of multi-stage training.

**RL Scalability.** We present the RL training reward and entropy curves in Figure 4 to study the impact of Entropulse on the extended RL training dynamics. After the first RL phase converges, we compare the second RL phase with and without Entropulse. To ensure a fair comparison, we reset the reference model in both scenarios.

The results demonstrate that incorporating Entropulse increases the model's entropy, thereby restoring its exploratory capacity. This enhanced exploration substantially scales the effective training steps, ultimately leading to improved overall performance.

## 4.4 CASE STUDY AND ERROR ANALYSIS

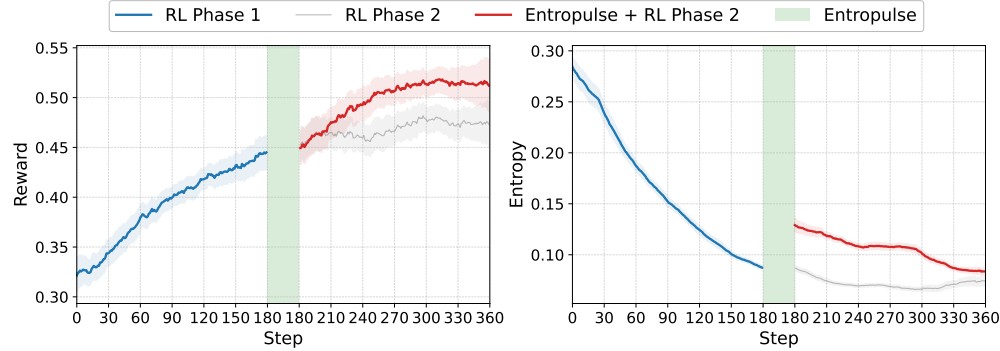

Figure 4: COMPUTERRL training curves of reward (left) and entropy (right) with 95% confidence intervals. The red line denotes the training with entropy recovery via Entropulse after the first RL stage, while the grey line denotes continued training with only reference resetting.

We conduct a case study in the desktop environment to identify potential avenues for system optimization. Although our model exhibits robust performance across most scenarios, several limitations have been identified. In particular, errors encountered during task execution can be categorized into four primary types: visual perception errors, multi-application coordination failures, operational illusions, and other errors. The distribution of these error types is presented in Figure 5.

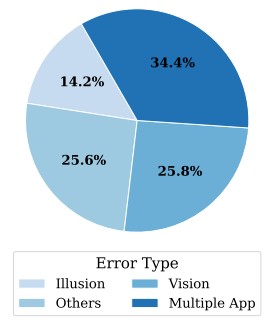

Figure 5: Error distribution.

## 4.5 EXPERIMENTS WITH DIFFERENT BASE MODELS

To further verify the effectiveness of our method, we conduct repetitive experiments with different base models (both text and multimodal), demonstrating the stability and superiority of our approach. The results are reported in Figure 6.

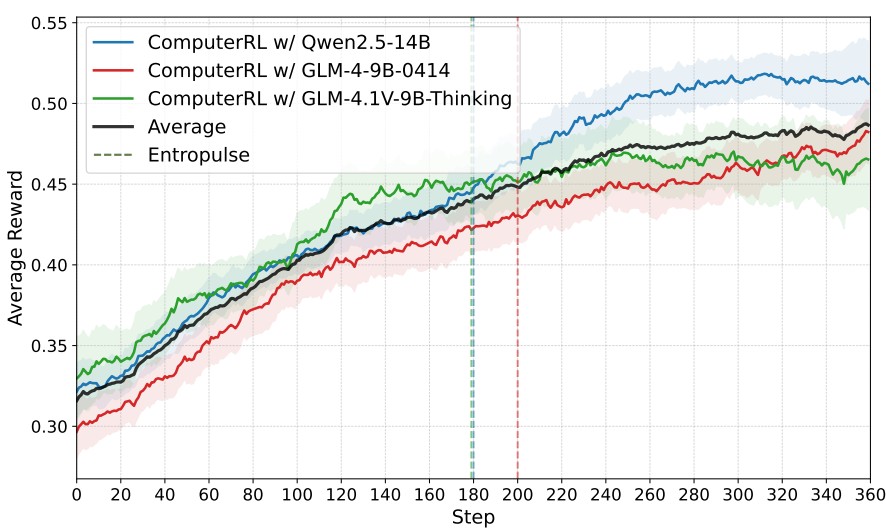

Figure 6: Repetitive experiments with different base models (95% CIs).

Appendix E presents more experimental results. Additional examples (including both good and bad) are provided in Appendix H to further illustrate the model's capabilities and limitations.

## 5    RELATED WORK

**Large Language Models.** LLMs, such as GPT (Achiam et al., 2023), Gemini (Team et al., 2023), Claude (Anthropic, 2023), Llama (Touvron et al., 2023a), GLM (Zeng et al., 2022; Du et al., 2022), Qwen (Team, 2024), and Deepseek (Liu et al., 2024a), have demonstrated remarkable capabilities in knowledge representation and language understanding, leading to diverse downstream applications. Vision-Language Models (VLMs) (Hong et al., 2023; 2025; Bai et al., 2023b; Hurst et al., 2024) further extend LLMs to multimodal inputs, enabling joint reasoning over text and images.

**Computer Use Agents.** CogAgent (Hong et al., 2023) introduces multimodal GUI understanding. AutoGLM (Liu et al., 2024b) decouples planning and grounding with online RL improvement. OS-Atlas (Wu et al., 2024b) proposes a foundational GUI action model. Aguvis (Xu et al., 2024) enables cross-platform interaction through visual training. PC-Agent-E (He et al., 2025) utilizes trajectory boosting for enhanced proficiency. UI-TARS (Qin et al., 2025) performs human-like GUI interactions from screenshots. Agent S2 (Agashe et al., 2025) integrates grounding with hierarchical reasoning. CUA (OpenAI, 2025) offers programmable desktop automation.

**Computer Use Benchmarks.** WebArena (Zhou et al., 2023) provides simulated websites for online interactions, but has limitations: discrepancies from real-world environments and a web-only focus. Similar issues exist in other web-focused benchmarks (Yao et al., 2022; Koh et al., 2024; Chezelles et al., 2024; Miyai et al., 2025). Software engineering benchmarks (Jimenez et al., 2023; Yang et al., 2024a; Li et al., 2024; Zan et al., 2025; Padigela et al., 2025) lack comprehensive desktop evaluation. OSWorld (Xie et al., 2024) addresses these gaps with 369 tasks with 134 evaluation functions. Windows Agent Arena (Bonatti et al., 2024) expands this with 150+ Windows-based tasks.

**RL and Entropy Management for LLMs.** PPO (Schulman et al., 2017) addresses instability in policy gradients for RL training. GRPO (Guo et al., 2025) extends PPO with group sampling and removes value updates. Maximum entropy RL (Haarnoja et al., 2018) and ensemble methods (Lee et al., 2021; De Paola et al., 2025) maintain diversity through regularization or multiple models. Recent work identifies entropy collapse as a critical challenge in LLM RL (Cui et al., 2025), with proposed solutions including DAPO (Yu et al., 2025) with adaptive clipping and token-level interventions (Hao et al., 2025). Entropulse takes a different approach by actively restoring collapsed entropy through targeted SFT training on diverse rollout data, achieving extended training.

## 6    CONCLUSION

In this work, we present COMPUTERRL, a novel computer use agent that integrates API-based and GUI-based actions with scalable RL training. Our experiments on OSWorld and OfficeWorld demonstrate superior performance compared to prior approaches, laying the groundwork for more capable autonomous computer use agents.

## ACKNOWLEDGMENT

This work was supported by Fundamental and Interdisciplinary Disciplines Breakthrough Plan of the Ministry of Education of China (No. JYB2025XDXM101), Natural Science Foundation of China (NSFC) 62425601 and 62495063, the New Cornerstone Science Foundation through the XPLORER PRIZE.

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

# A    API DEVELOPMENT WORKFLOW

In this section, we detail the methodology for leveraging LLMs to automate API construction. We propose a semi-automated workflow wherein users need only supply exemplar tasks performed within the target application; the LLM then autonomously generates both the necessary API code and corresponding test cases. The workflow comprises three primary stages: requirement analysis, API implementation, and test case generation.

**Requirement Analysis** During the requirements analysis phase, users provide a set of task examples related to the target application as input. The workflow leverages the LLM to analyze these task instances, extracting the essential functionalities required for task completion. It then compares these requirements against the existing API interface definitions to identify potential gaps. If uncovered functionalities are detected, the system automatically generates new API interfaces along with their corresponding parameter specifications.

Notably, we limit the generated interfaces to encapsulate only general-purpose functionalities, thereby avoiding excessive complexity and the proliferation of APIs. This design choice mitigates implementation difficulty and reduces the adaptation burden on the agent.

**API Implementation** Upon obtaining the interface definitions, the workflow systematically iterates over each interface and its associated parameters. For each specification, it leverages the designated Python libraries of the target application to implement the corresponding API functionalities. Additionally, the workflow incorporates error-handling mechanisms and logging to facilitate human debugging and maintenance. This automated approach not only streamlines API development but also enhances consistency and reusability across different application contexts.

**Test Case Generation** Following the implementation of API functionalities, the workflow conducts fundamental unit testing to ensure the correctness and robustness of each API. Specifically, the testing process verifies: (1) whether the API can be invoked without runtime errors, and (2) whether the API returns correct results across a range of parameter inputs. For API implementations that fail these tests, the workflow provides detailed error feedback to the API implementation module, which then autonomously attempts corrections until the APIs pass all tests.

This automated methodology substantially lowers the manual effort required, enabling the creation of application-specific API sets with minimal human intervention. As a result, the barrier for users to develop APIs for diverse applications is significantly reduced. We have developed API sets for multiple commonly used default applications in Ubuntu and integrated them into our Ubuntu virtual machine environment.

# B ACTION SPACE

Our action space for GLM-COMPUTERRL is shown in Table 4, and our API number for each application is in Table 5.

Table 4: Action space of GLM-COMPUTERRL

| Function | Description |
|----------|-------------|
| open_app(app_name) | Open specified application (e.g., Chrome, Terminal). |
| click(coordinates, num_clicks,button_type) | Click at coordinates [x, y] with the specified mouse button and number of clicks. |
| type(coordinates, text, overwrite, enter) | Type text at coordinates; optionally overwrite existing content and/or press Enter. |
| drag_and_drop(drag_from, drop_on) | Drag from $[x_1, y_1]$ and drop onto $[x_2, y_2]$. |
| scroll(coordinates, direction) | Scroll at coordinates in direction (up / down). |
| switch_window(window_id) | Switch focus to the window with given ID. |
| hotkey(keys) | Press a key combination (e.g., [ctrl, c]). |
| quote(content) | Record content for memory. |
| wait() | Pause execution temporarily. |
| exit(success) | Terminate task with success (True) or failure (False). |

Table 5: Statistics of the number of available APIs per application

| Application | Number of APIs |
|-------------|---------------:|
| Code | 12 |
| Chrome | 11 |
| LibreOffice Calc | 27 |
| LibreOffice Impress | 22 |
| LibreOffice Writer | 19 |
| VLC | 12 |
| Total | 103 |

## C   PROMPT FORMULATION FOR GLM-COMPUTERRL

The design of the observation space is pivotal, as it directly constrains the upper bound of the agent's performance. In this section, we detail the integration of the API set with GUI operations, alongside the incorporation of contextual information from the desktop environment, to systematically construct both the agent's observation space and action space. This unified framework ensures that the designed observation and action spaces capture the complexity of real-world tasks, providing a solid foundation for robust agent learning and generalization.

**Action space formulation** The integration of a large number of APIs with GUI operations, while ensuring effective agent interaction, remains a significant challenge. In practice, we mitigate this complexity by dynamically detecting the currently active application to infer potentially relevant APIs, thereby reducing the number of available APIs and lowering the agent's adaptation overhead. Furthermore, we use Python classes and descriptive docstrings to delineate each operation type, ensuring they are clearly interpretable by most LLMs. This object-oriented strategy enhances the model's understanding and precision in performing operations. These classes are provided to the agent via the system prompt, enabling interaction through Python function calls. This design facilitates rapid agent adaptation and efficient generalization of operations across diverse applications. Additionally, the agent's output format is standardized in the system prompt, which encourages the agent to interleave reasoning and action execution. This approach promotes enhanced planning and reflective capabilities within the agent, thus improving its overall performance in complex task execution scenarios.

**Observation formulation** To facilitate the effective perception and manipulation of GUIs by the model, we leverage the Python Accessibility Toolkit Service Provider Interface (pyatspi) to extract comprehensive attributes of desktop elements systematically. Each GUI element encompasses the element's semantic type, visible text content, precise screen coordinates, and spatial dimensions. This structured representation enables the LLM agent to parse, ground, and reason over the GUI in a manner analogous to human users.

We present the element format of the environment a11y tree in our observation space as follows:

```
tag      text       position (center x & y)      size (w & h)
```

The `tag` is the XML tag of the element, such as `div` or `button`. The `text` is the text content of the element, which can be empty for elements that do not have text. The `position` is represented by the center coordinates (x, y) of the element, and the `size` is represented by its width (w) and height (h).

For the multimodal model, the a11y tree is removed from the input. Instead, we capture the GUI screenshot at a resolution of $1920 \times 1080$ pixels (1080p) and subsequently resize it to $1280 \times 720$ pixels (720p), which serves as the input representation of the desktop environment.

Beyond the extraction of individual GUI components, we augment the input space with rich contextual metadata to provide a holistic depiction of the agent's operational environment. Specifically, we provide a comprehensive enumeration of open desktop windows, including their hierarchical relationships, as well as the name and additional information of the currently focused application. To promote consistent and adaptive behavior, we also deliver feedback from the most recent GUI action or tool call, which may include environmental status updates, confirmations, or error signals.

The app format of the observation space is as follows:

```
Window ID     App Name      Title
```

The `Window ID` is the unique identifier of the application window, `App Name` is the name of the application, and `Title` is the title of the application window.

**History formulation** Given the extensive length of GUI observations and the inherent constraints imposed by the model's context window, it is necessary to efficiently manage the input history across multiple interaction rounds. For each interaction, we omit redundant and detailed interface information while preserving the complete sequence of the agent's reasoning process, actions taken, and the corresponding operation feedback. This approach ensures the retention of the essential operational trajectory, thereby maximizing the informativeness of the historical context while maintaining compatibility with the model's capacity limitations.

Collectively, the components above constitute the observation space and action space of our agent. This representation not only enhances the agent's environmental cognition but also enables better strategies for long-horizon planning and reasoning. As a result, the agent is better equipped to execute complex, multi-step tasks across diverse applications in the desktop environment.

Below is our detailed prompt organization for GLM-COMPUTERRL:

```
You are an agent which follow my instruction and perform desktop computer
    tasks as instructed.
You have good knowledge of computer and good internet connection and
    assume your code will run on a computer for controlling the mouse and
    keyboard.
For each step, you will get an observation of the desktop by 1)
    screenshot; 2) current application name; 3) accessibility tree, which
     is based on AT-SPI library; 4) application info; 5) last action
    result.
You should first generate a plan for completing the task, confirm the
    previous results, reflect on the current status, then generate
    operations to complete the task in python-style pseudo code using the
     predefined functions.

Your output should STRICTLY follow the format:
<think>
{**YOUR-PLAN-AND-THINKING**}
</think>
'''python
{**ONE-LINE-OF-CODE**}
'''

You will be provided access to the following methods to interact with the
     UI:
    1. class Agent, a grounding agent which provides basic action space
        to interact with desktop.
    2. class {tool_class_name}, which provides tools to interact with the
         current application {app_name}.

Here are the defination of the classes:
'''python
{class_content}
'''

* Note:
- Your code should be wrapped in '''python''', and your plan and thinking
     should be wrapped in <think></think>.
- Only **ONE-LINE-OF-CODE** at a time.
- Each code block is context independent, and variables from the previous
     round cannot be used in the next round.
- Do not put anything other than python code in '''python'''.
- You **can only use the above methods to interact with the UI**, do not
    invent new methods.
- Return with 'Agent.exit(success=True)' immediately after the task is
    completed.
- If you think cannot complete the task, **DO NOT keep repeating actions,
     just return with 'Agent.exit(success=False)'.**
- The computer's environment is Linux, e.g., Desktop path is '/home/user/
    Desktop'
- My computer's password is 'password', feel free to use it when you need
     sudo rights

**IMPORTANT** You are asked to complete the following task: {instruction}
```

Below is our history and input prompt for GLM-COMPUTERRL:

```
<|user|>
**Environment State (Omitted)**
```

```
<|assistant|>
<think>
{round0_thinking}
</think>
'''python
{round0_operation}
'''

<|user|>
**Environment State (Omitted)**
Previous Action Result: {round0_operation_feedback}

<|assistant|>
<think>
{round1_thinking}
</think>
'''python
{round1_operation}
'''

<|user|>
**Environment State (Omitted)**
Previous Action Result: {round1_operation_feedback}

<|assistant|>
<think>
{round2_thinking}
</think>
'''python
{round2_operation}
'''
...

<|user|>
{screenshot_for_multimodal}
* Apps: {all_apps}

* Current App: {cur_window_id}

* A11y Tree: {a11y_tree_for_text}

* App Info: {app_info}

* Previous Action Result: {operation_feedback}
```

## D  TRAINING & HARDWARE DETAILS

### D.1  TRAINING PROCESS & HYPERPARAMETER SETTINGS

During the behavior cloning stage, we construct approximately 8,000 tasks through manual annotation and data augmentation. We employ multiple advanced models to generate diverse samples for each task, and subsequently apply the eval function to filter out successful trajectories. This process yields a high-quality BC dataset comprising roughly 180,000 steps, which is then used for SFT training. We employ a 16-node computing cluster for fine-tuning, with a maximum learning rate set to $1 \times 10^{-5}$, a sequence length of 32,768 tokens, and a global batch size of 256, over a total of three training epochs.

In the RL stage, the key training hyperparameters are summarized in Table 6. We initially train the BC policy (using the 1-epoch checkpoint for diversity) for 180 steps, after which performance improvements began to plateau. At this point, we collect rollouts during RL, perform task-level random selection, and curate approximately 130,000 additional steps of data for Entropulse training. The hyperparameters in this phase are identical to those used in the BC stage, except for a reduced learning rate of $5 \times 10^{-6}$. RL training is then resumed until a total of 360 steps have been reached.

### D.2  TRAINING CLUSTER CONFIGURATION

Our training infrastructure consists of a high-performance GPU cluster. The complete specifications, including GPU, CPU, cache, memory, and network configuration, are detailed in Table 7. Our training pipeline requires at least **4 GPU nodes** to run distributed RL training.

### D.3  ENVIRONMENT CLUSTER CONFIGURATION

For running distributed RL environments, we employ a dedicated compute cluster with 7 nodes. The complete specifications are shown in Table 8. In our empirically validated deployment:

- Each GPU achieves optimal utilization when paired with approximately **80 rollouts**.
- Each environment server can reliably host **200 concurrently running virtual environments**.

This ratio maintains equilibrium between GPU computation and environment sampling, minimizing idle computational resources.

### D.4  VIRTUAL ENVIRONMENT INSTANCE CONFIGURATION

Each RL task is executed within a dedicated virtual machine instance. The specifications are detailed in Table 9.

**Note:** Each virtual environment instance runs an independent Ubuntu 20.04 desktop environment for executing GUI-based tasks. The lightweight resource configuration (2 cores/4GB) ensures high concurrency under limited hardware resources, supporting the environment parallelism required for large-scale distributed RL training.

### D.5  TRAINING DURATION AND FLOPS STATISTICS

Table 10 presents the complete training time and FLOPs statistics for the multimodal training.

Table 6: Training configuration for RL training of GLM-COMPUTERRL.

| Category | Parameter (Value) |
|---|---|
| **Data** | |
| Task type | Multi-turn chat |
| Max prompt length | 63,488 tokens |
| Max response length | 2,048 tokens |
| Train batch size | 32 |
| Responses per prompt | 16 |
| Concurrency | 1024 |
| Shuffle | True |
| Seed | 42 |
| **Actor** | |
| Exchange size | $1 \times 10^{10}$ |
| Gradient checkpointing | Enabled |
| Strategy | FSDP |
| FSDP offloading | Param + Optimizer |
| Sequence parallel size | 2 |
| Max tokens / GPU | 32,768 |
| Precision dtype | bfloat16 |
| **Algorithm** | |
| Advantage estimator | GRPO |
| Discount factor $\gamma$ | 1.0 |
| GAE parameter $\lambda$ | 1.0 |
| Use remove padding | True |
| Use dynamic bsz | True |
| Mini-batch size | 32,768 |
| Micro-batch size / GPU | 1 |
| Logprob micro-batch size / GPU | 1 |
| KL loss | Enabled (*low_var_kl*), coef = 0.0003 |
| Entropy coefficient | 0.0 |
| Clip ratio | 0.2 |
| **Optimizer** | |
| Actor learning rate | $1 \times 10^{-6}$ |
| LR warmup steps ratio | 0.0 |
| Warmup style | constant |
| Gradient clipping | 1.0 |
| Save frequency | 25 |
| **Rollout** | |
| Enable chunked prefill | True |
| Max new tokens (generation) | 2,048 |
| Do sample | True |
| Sampling temperature | 0.8 |
| Max turns | 30 |
| GPU memory utilization | 0.7 |
| Pools rollout | 2 |
| Pools other | 6 |

Table 7: Training Cluster Specifications

| Configuration | Specification |
|---|---|
| *Cluster Overview* | |
| Cluster Size | 16 nodes |
| *GPU Configuration* | |
| GPU Type | NVIDIA H800 |
| GPUs per Node | 8 |
| Total GPUs | 128 (16 nodes × 8 GPUs) |
| *CPU Configuration* | |
| CPU Model | Intel Xeon Gold 6430 |
| Architecture | x86_64 |
| Physical Sockets | 2 |
| Cores per Socket | 32 |
| Total Cores | 64 |
| Threads | 64 (1 thread/core) |
| Base Frequency | 2.1 GHz |
| Minimum Frequency | 800 MHz |
| Instruction Set Extensions | AVX-512, AVX512_FP16, AMX (INT8/BF16/Tile) |
| *Cache Configuration* | |
| L1 Data Cache | 3 MiB (64 instances) |
| L1 Instruction Cache | 2 MiB (64 instances) |
| L2 Cache | 128 MiB (64 instances) |
| L3 Cache | 120 MiB (2 instances) |
| *Memory Configuration* | |
| Total Memory Capacity | 2.0 TiB |
| Available Memory | 1.9 TiB |
| NUMA Nodes | 2 |
| NUMA Node 0 CPUs | 0-31 |
| NUMA Node 1 CPUs | 32-63 |
| Swap | Disabled (0 B) |
| *Network Configuration* | |
| Interconnect | InfiniBand/High-speed Ethernet |
| Address Width | Physical 46-bit, Virtual 57-bit |

Table 8: Environment Cluster Specifications

| Configuration | Specification |
|---|---|
| *Cluster Overview* | |
| Cluster Size | 7 nodes |
| Total Cluster Memory | ∼7.7 TiB |
| *CPU Configuration* | |
| CPU Model | Intel Xeon 6986P-C (Granite Rapids) |
| Architecture | x86_64 |
| Physical Sockets | 1 |
| Cores per Socket | 120 |
| Total Cores | 120 |
| Threads | 240 (2 threads/core, hyper-threading enabled) |
| Base Frequency | 3.3 GHz |
| Max Turbo Frequency | 3.9 GHz |
| Minimum Frequency | 800 MHz |
| Instruction Set Extensions | AVX-512, AVX512_BF16, AMX, SHA-NI |
| *Cache Configuration* | |
| L1 Data Cache | 5.6 MiB |
| L1 Instruction Cache | 7.5 MiB |
| L2 Cache | 240 MiB |
| L3 Cache | 504 MiB |
| *Memory Configuration* | |
| Memory per Node | 1.1 TiB |
| Available Memory | 949 GiB |
| NUMA Nodes | 3 |
| NUMA Node 0 CPUs | 0-39, 120-159 |
| NUMA Node 1 CPUs | 40-79, 160-199 |
| NUMA Node 2 CPUs | 80-119, 200-239 |
| Swap | Disabled (0 B) |
| *Virtualization and Features* | |
| Virtualization Technology | Intel VT-x, EPT, VPID |
| Security Features | Enhanced IBRS, IBPB, Spectre/Meltdown mitigations |
| Cryptographic Acceleration | AES-NI, SHA-NI, AVX512_VAES |
| AI Acceleration | AMX, AVX512_BF16, AVX512_VNNI |
| Address Width | Physical 52-bit, Virtual 57-bit |

Table 9: Virtual Environment Instance Specifications

| Configuration | Specification |
|---|---|
| Operating System | Ubuntu 20.04 LTS |
| vCPU Cores | 2 |
| Memory Allocation | 4 GB |
| Runtime Average Bandwidth | 0.4 Mbps |
| Virtualization Platform | KVM/QEMU |

Table 10: Training Time and FLOPs Statistics (Multimodal)

| Training Stage | Duration (hours) | Total FLOPs |
|---|---|---|
| SFT (Behavior Cloning) | 16 | $1.67 \times 10^{16}$ |
| SFT (Entropulse) | 11 | $1.22 \times 10^{16}$ |
| RL (Two-stage) | 58 | $3.21 \times 10^{17}$ (estimated) |

### D.6 Key Performance Trade-offs and Bottlenecks

Our observations identify two principal hyperparameters that significantly influence the balance between training efficiency and convergence stability.

#### 1. Responses per Prompt (RpP) – Exploration Upper Bound vs. Sampling Efficiency

**Role.** RpP defines the breadth of the search space in Best-of-N (BoN) sampling.

**Trade-off:**

- A **higher RpP** broadens exploration, increasing the likelihood of discovering high-quality trajectories, but sampling latency grows roughly linearly.
- A **lower RpP** yields faster sampling but may omit promising solutions, constraining exploration scope.

**Bottleneck.** Excessively large RpP values incur substantial sampling overhead with diminishing marginal gains.

#### 2. Batch Size (B) – Training Stability vs. Iteration Throughput

**Role.** B specifies the number of samples processed in each gradient update.

**Trade-off:**

- A **larger B** improves gradient estimation accuracy and stabilizes training, but extends iteration time.
- A **smaller B** accelerates iterations but introduces higher gradient variance, potentially destabilizing convergence.

**Bottleneck.** Too small B values cause pronounced oscillations in the training curve, while too large values extend iteration time.

**Optimal Configuration: RpP = 16, B = 32.** Systematic experimentation confirms that RpP $= 16$ and $B = 32$ represent an optimal balance across competing objectives:

- **Exploration Adequacy** – RpP = 16 affords sufficient BoN sampling scope to cover the majority of feasible solution trajectories.
- **Training Stability** – B = 32 maintains variance in gradient estimates within acceptable bounds, promoting smooth convergence.
- **Resource Efficiency** – This configuration ensures balanced utilization of both GPU and environment clusters, avoiding throughput bottlenecks.
- **Performance Outcome** – Using this configuration, we achieved the reported final performance, outperforming other settings in the efficiency–accuracy trade-off.

### D.7 Additional Observed Bottlenecks

#### 1. Environment Heterogeneity

**Issue.** Significant variance in task execution time results in some GPUs waiting for slower environments to complete.

**Mitigation.** An asynchronous rollout collection mechanism allows fast environments to submit results without delay.

#### 2. Inter-cluster Network Bandwidth

**Issue.** High concurrency in environment simulation can saturate network bandwidth due to frequent transmission of screenshots and state data, occasionally causing Docker network stalls.

**Mitigation.** Employing image compression reduces network load; optimizing Docker networking decreases virtual NIC overhead.

3. INTERNET BANDWIDTH CONSTRAINTS

**Issue.** Large-scale simultaneous environment instances can generate excessive external network traffic.

**Mitigation.** Packet-level traffic analysis enables elimination of unnecessary transmissions; constructing an IP proxy pool mitigates service blocking risks.

# E   ADDITIONAL EXPERIMENTAL INDICATORS

To more comprehensively validate the effectiveness of our method, we report detailed experimental indicators to support our conclusions, as shown in Figure 7. These indicators include Average Reward, Entropy Loss, KL Loss, PPO KL, Average Margin, BoN Reward, Average Turns, and Response Length. Based on these metrics, we make the following observations:

- Entropulse effectively increases the stochasticity of the policy, leading to a substantial improvement in BoN after activation. This, in turn, drives the growth of the margin and enables the policy to continue learning and improving.
- After applying Entropulse, the response patterns of the policy (including response length and number of dialogue turns) become closer to those before the first-stage RL training (i.e., shorter), while maintaining comparable scores. This indicates that Entropulse helps the policy discover better solutions along shorter trajectories, thereby suppressing excessive reasoning and redundant steps.
- After resetting the reference model, the KL Loss is also reset, allowing the policy to explore a larger space relative to the new reference. This prevents the policy from being overly constrained by its previous strategy.

# F   HUMAN ANNOTATION PROTOCOL

Our annotation process involves ten trained annotators with master's degrees, who are recruited and compensated in compliance with local labor laws and regulations. Annotators are provided with clear written guidelines to ensure consistency and accuracy, as outlined in our annotation protocol (see Figures 8 and 9). All tasks are designed to avoid sensitive personal data, and all annotated content is in English with no identifying information. The process includes task expansion—transforming generalized instructions into explicit, executable tasks—and strict result verification to minimize errors. Quality control measures include verification passes and clear formatting rules to improve annotation reliability. No annotator is exposed to harmful, discriminatory, or unsafe content during the process, and all work adheres to the Code of Ethics regarding fairness, privacy, and legal compliance.

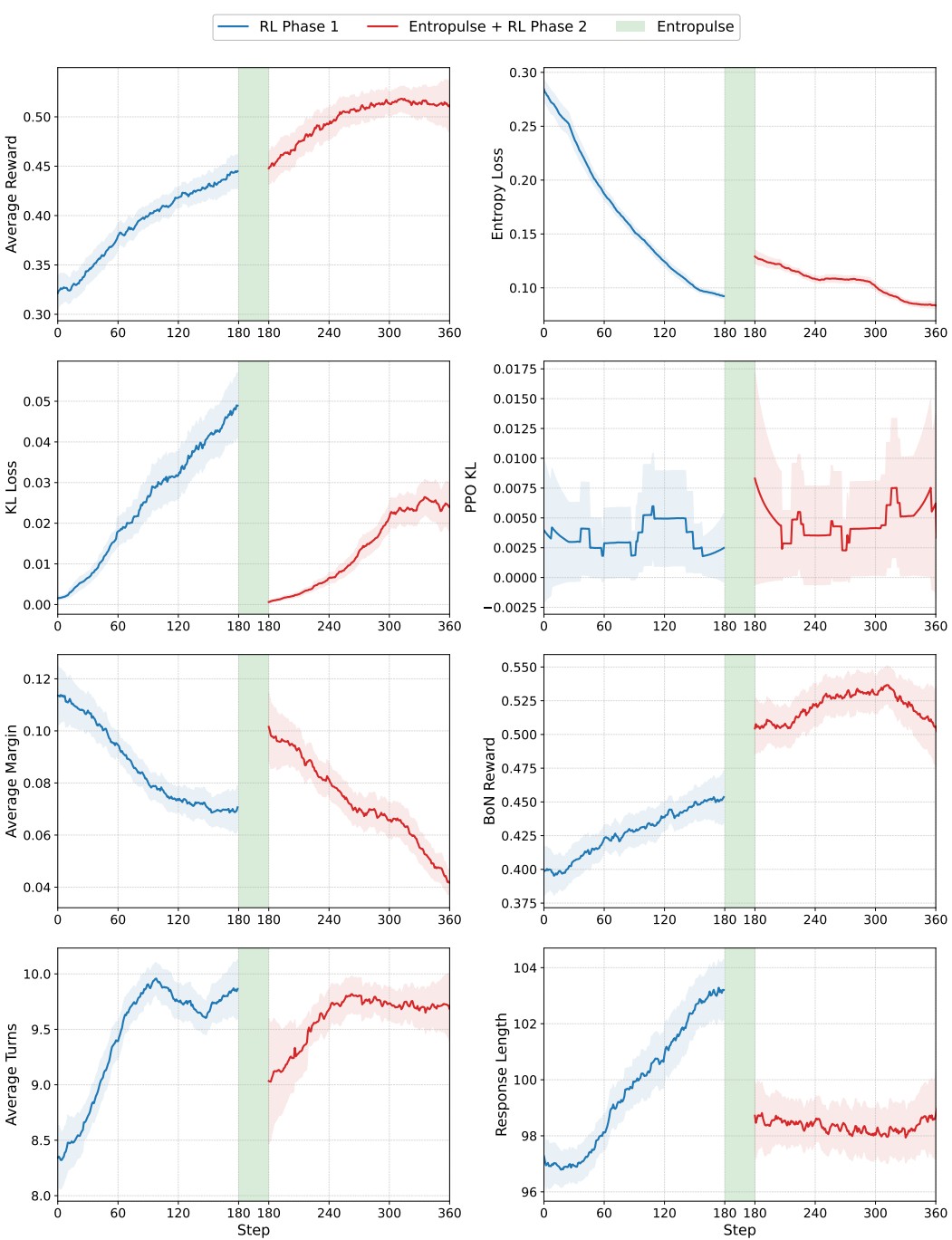

Figure 7: Detailed experimental indicators for COMPUTERRL (95% CIs).

# Annotation Guidelines

## I. Project Background

Currently, when agents perform different application tasks, various execution issues may arise. To address this problem, the core of this annotation task is:

1. First, **expand the user's seed instruction** into more executable instructions.
2. Then, **conduct strict result verification** after each task execution to ensure the accuracy of the execution chain.

## II. Data Description

### Phase 1: Task Expansion

**Fields to be completed** (No need to execute in a virtual machine, only need to fill in):

1. **Expanded Task**
   - Modify the generalized task description into an **independent, executable** task.
   - **No pronominal references** allowed.
   - All operations must explicitly specify that they are performed **on a file located under** `/home/user/work/` (the prefix `/home/user/work/` is fixed).
   - Instructions must be as clear and executable as possible.
     - **Prohibited example**: "Help me find some good movies" (too vague).
   - You may appropriately **reduce the difficulty** of the task, for example:
     - If the original task is "Collect data from 2013–2020," you may adjust it to "Collect data from 2013–2015" (simplify content preparation for Phase 2).
2. **Notes** – Provide guidance for Phase 2 (described in Chinese for convenience).
   - Example:

     > "Prepare a `test.docx` file containing multiple sentences, where the first letter of each sentence is lowercase. Verify using `compare_with_golden(result_path, golden_path)`. The result passes if the first English letter of each sentence is capitalized, other characters remain unchanged, and the content is exactly the same as the golden file."

   - You must specify:
     a. **What type of file** to prepare
     b. **The file content format**

Figure 8: Document for Human Annotation (1 / 2)

      c. **Which function** to use for verification

      d. **Which fields** to check during verification

### Phase 2: Virtual Machine Function Verification & Field Completion

**Fields to be completed**:

1. **Target File**
   - All files in the task that require model operations (unprocessed files).
   - For `compare_with_golden` (only supports `xlsx`, `docx`, `csv`, `txt`):
     - If the task involves other file types or functions, these can be omitted.
2. **Detection Target**
   - The name(s) of the file(s) that need to be compared.
3. **Golden File** *(only filled when the verification function is `compare_with_golden`)*
   - Name of the standard answer file used for comparison.
4. **check_list** *(required if the Notes or task description explicitly specifies formatting)*:
   - Optional if content comparison is default.
   - Single/multiple choice:
     - `font`: Compare font name, color, size, bold, italic, etc.
     - `fill`: Compare cell fill color.
     - `alignment`: Compare alignment (e.g., center).
     - `para_format`: Compare paragraph formatting (alignment, line spacing).
     - `table`: Compare text in inserted tables (format comparison not currently supported).
5. **Files**
   - All files required for the task.
   - You must create your own file names (arbitrary), but the file names **must be consistent across all fields** (important).

## III. Annotation Notes

1. **Focus on evaluation** — ensure the model's required operations are properly executed; avoid cases where detection passes without actual modifications.
2. **All data must be in English**, ensure logical correctness after translation.
3. For manually typed scenarios, verify multiple times to minimize error rate:
   - Especially check IDs and file paths.
   - OCR (from screenshots) is recommended for accuracy.
4. Report issues promptly in the annotation group to avoid unnecessary rework.
5. For color-related tasks, use **red, yellow, blue, green**; avoid visually similar colors.
6. For downloading plugins or apps — **assume they are already installed**.

Figure 9: Document for Human Annotation (2 / 2)

## G    FUTURE DIRECTION

While our advancements with COMPUTERRL mark a significant leap forward in intelligent desktop automation, we see this work as a foundation for a radical transformation in human-computer interaction. Unleashing the full potential of autonomous agents on the desktop frontier demands reimagining long-standing paradigms across several axes.

### G.1    TOWARDS ROBUST PERFORMANCE

GLM-COMPUTERRL has demonstrated remarkable proficiency across a spectrum of desktop tasks. However, genuine universality requires transcending current boundaries in coverage and generalization. Real-world digital environments are characterized by continual flux and heterogeneity, encompassing unfamiliar applications, emergent workflows, and rare edge cases that lie beyond the scope of existing datasets. A next-generation agent must dynamically adapt to shifting GUIs, unpredictable pop-ups, and entirely novel interfaces. To this end, we are re-architecting data pipelines to facilitate exponential expansion in training diversity and pioneering infrastructure to distill knowledge from real-world user interactions at scale continuously.

### G.2    BREAKTHROUGHS IN LONG-HORIZON AUTONOMY

Envisioning the autonomous desktop assistant as an always-available cognitive collaborator necessitates mastering sustained, long-duration workflows. While current solutions excel at bounded, atomic tasks, they fall short of orchestrating complex, multi-step objectives over extended horizons. Our ambition is to endow the agent with hierarchical planning capabilities, allowing it to reason, learn, and revise strategies dynamically across arbitrarily long task sequences. Realizing this vision will catalyze a paradigm shift: automating not just discrete operations, but entire workstreams and creative processes end-to-end, fundamentally reshaping the productivity landscape for the cloud-native era.

### G.3    FOUNDATIONS FOR SAFE AND ALIGNED AUTONOMY

Autonomous control over desktop platforms raises profound questions about safety, trustworthiness, and user agency. The margin for error narrows dramatically when agents are empowered to modify files, access sensitive data, or execute unbounded actions. Mitigating these risks requires a rigorous and principled approach to safe behavior and alignment. Our roadmap includes architecting granular permissioning frameworks, embedding robust pre-action validation, and multi-stage approval protocols. Ultimately, we aim to establish safety standards and best practices that can serve as foundational infrastructure, not just for our agent but for the future ecosystem of intelligent digital collaborators.

## H    DEMONSTRATION

This section presents examples drawn from a variety of application scenarios, including the initial four positive cases and the final two negative ones. Each example illustrates our API-GUI operational paradigm, which addresses the diverse challenges and requirements that arise within different applications.

(A): Copy an image from a .xcf file on the Desktop and paste it into a LibreOffice Writer document, then save it as 'image.docx' on the Desktop.

(B): Use the `sar` command from the `sysstat` toolkit to monitor Ubuntu system resources every second for 30 seconds, and save the results to 'System_Resources_Report.txt' on the Desktop.

(C): Create a table with 'Month' and 'Total' headers in a new sheet to show the total sales for all months.

(D): Convert all uppercase text to lowercase in a document for consistency.

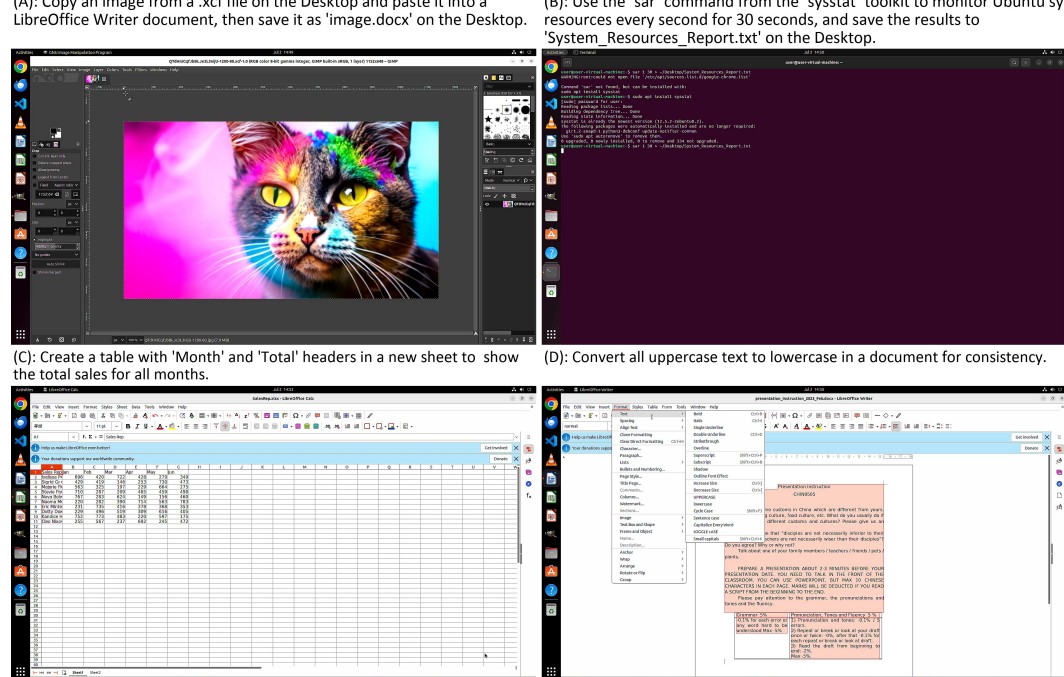

Figure 10: Examples of GLM-COMPUTERRL's execution on four user tasks, including image processing between GIMP and LibreOffice Writer, monitoring system resource usage in Terminal, table calculation in LibreOffice Calc, and document formatting in LibreOffice Writer.

### H.1    CREATE MONTHLY SALES SUMMARY TABLE

The targeted task to be executed is "Create a table with two headers ('Month' and 'Total') in a new sheet to show the total sales for all months." The actual execution steps can be summarized as follows:

- Step1: Switch to 'Sheet2'
- Step2: Set cell 'A1' as "Month"
- Step3: Set cell 'B1' as "Total"
- Step4: Get monthly sales data from 'Sheet1'
- Step5: Switch to 'Sheet1'
- Step6: Switch to 'Sheet2'
- Step7: Set month names in column A
- Step8: Set total formula for Jan in 'B2'
- Step9: Set total formula for Feb in 'B3'
- Step10: Set total formula for Mar in 'B4'
- Step11: Set total formula for Apr in 'B5'
- Step12: Set total formula for May in 'B6'
- Step13: Set total formula for Jun in 'B7'

- Step14: Exit after completion

As Figure 11,12 shows, a summary table is created successfully.

## H.2 CONVERT UPPERCASE TEXT TO LOWERCASE IN DOCUMENT

The targeted task to be executed is "Convert all uppercase text to lowercase in the document for consistent text presentation." The actual execution steps can be summarized as follows:

- Step1: Use the text case conversion tool to change all text to lowercase.
- Step2: Click the formatting menu.
- Step3: Select the specific formatting option.
- Step4: Open the relevant submenu.
- Step5: Select all text in the document.
- Step6: Click the formatting menu again.
- Step7: Select the specific formatting option again.
- Step8: Open the relevant submenu again to apply the changes.
- Step9: Exit after finishing the operation.

As Figure 13 shows, all uppercase text has been successfully converted to lowercase, ensuring a uniform and polished document appearance.

## H.3 GENERATE SYSTEM RESOURCE MONITORING REPORT

The targeted task to be executed is "Use the 'sar' command in the 'sysstat' toolkit to monitor system activity and output the results to 'System_Resources_Report.txt' under Desktop." The actual execution steps can be summarized as follows:

- Step1: Open the Terminal application.
- Step2: Type the command sar 1 30 > /Desktop/System_Resources_Report.txt.
- Step3: Execute sudo apt install sysstat to install the toolkit.
- Step4: Enter the password if prompted.
- Step5: Re-type the sar command to start monitoring.
- Step6: Wait for 30 seconds while system activity is being recorded.
- Step7: Open the Files application.
- Step8: Navigate to and click on the report file under Desktop.
- Step9: Exit after confirming the report has been generated.

As Figure 14 shows, a system resource report file was generated successfully on the Desktop after monitoring, fulfilling the target task.

## H.4 COPY IMAGE FROM .xcf FILE TO LIBREOFFICE WRITER AND SAVE

The targeted task to be executed is "Copy the image from an .xcf file located on the Desktop and paste it into a LibreOffice Writer document, then save the document as image.docx on the Desktop." The actual execution steps can be summarized as follows:

- Step1: Open the Files application.
- Step2: Navigate to the Desktop folder.
- Step3: Double-click to open the .xcf image file.
- Step4: Copy the image in the image editor.
- Step5: Open LibreOffice Writer.

- Step6: Paste the copied image into the Writer document.
- Step7: Save the document.
- Step8: Enter the filename as `image.docx` and select the Desktop as the save location.
- Step9: Navigate to the Desktop folder.
- Step10: Confirm and click save.

As Figure 15 shows, the image from the `.xcf` file was successfully pasted into a LibreOffice Writer document and saved as `image.docx` on the Desktop, completing the task.

## H.5 QUESTION MISUNDERSTANDING ERROR

The targeted task to be executed is "Append 
 to the end of each line in '1\n2\n3' and save in output.txt." The actual execution steps can be summarized as follows:

- Step1: Type the command `echo -e '1\n2\n3'` in the terminal.
- Step2: Exit after execution.

As Figure 16 shows, the agent misunderstood the requirement and only echoed the content without appending 
 to each line or saving it into output.txt. This results in a task failure due to a misunderstanding of the question.

## H.6 CLICK OPERATION ERROR

The targeted task to be executed is "Please help change GIMP's theme from dark to light." The actual execution steps can be summarized as follows:

- Step1: Click on the menu in GIMP.
- Step2: Attempt to click the Preferences option.
- Step3: Repeat the click on Preferences.
- Step4: Try to use the shortcut `Shift+Ctrl+P` to open Preferences.
- Step5: Exit without successfully changing the theme.

As Figure 17 shows, the theme remains dark, indicating that the agent failed to change GIMP's theme to light due to incorrect click operations.

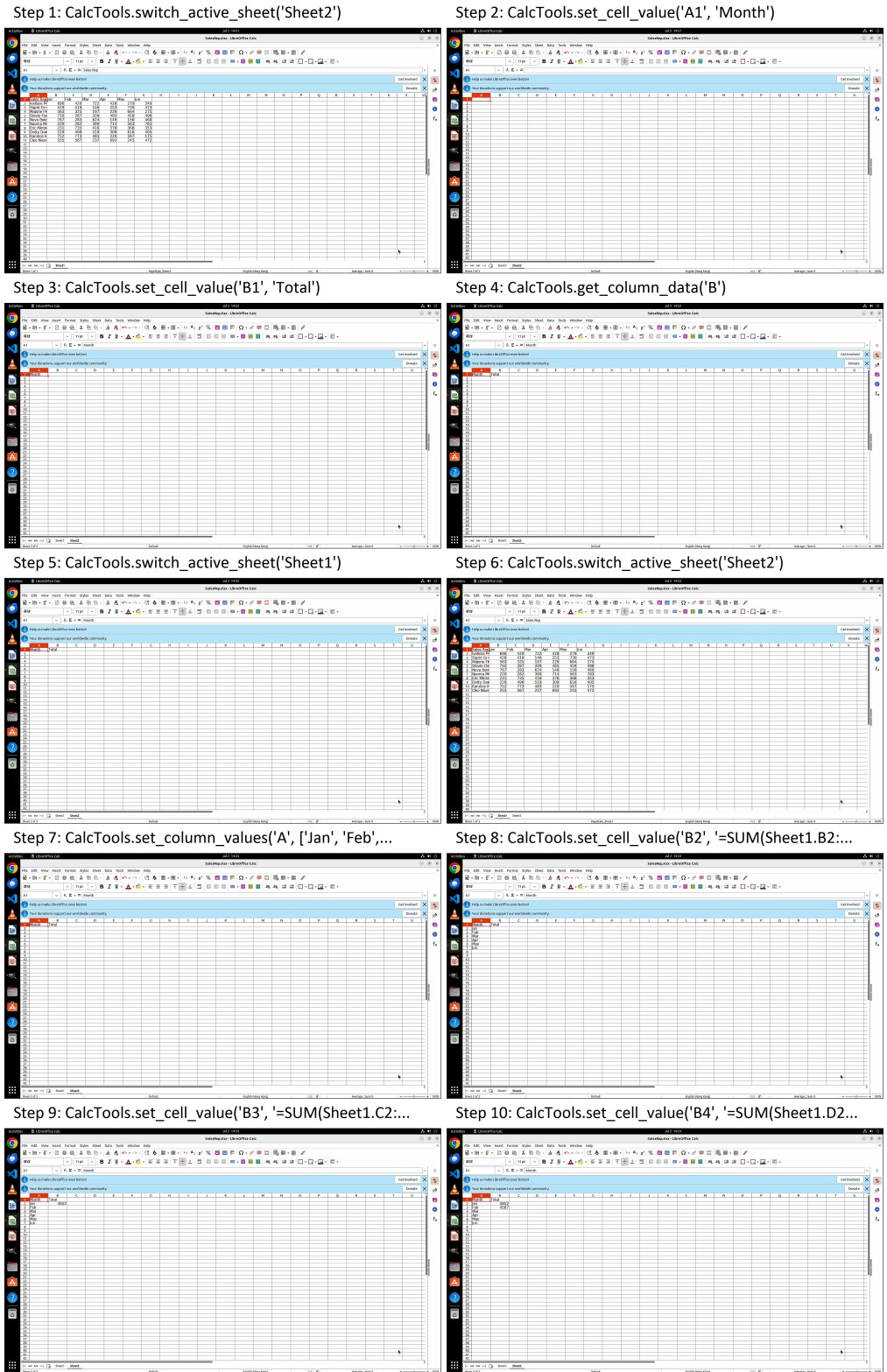

Figure 11: Task (Step 1-10): Create a table with two headers ("Month" and "Total") in a new sheet to show the total sales for all months.

Step 11: CalcTools.set_cell_value('B5', '=SUM(Sheet1.E2...    Step 12: CalcTools.set_cell_value('B6', '=SUM(Sheet1.F2...

Step 13: CalcTools.set_cell_value('B7', '=SUM(Sheet1.G2...    Step 14: Agent.exit(success=True))

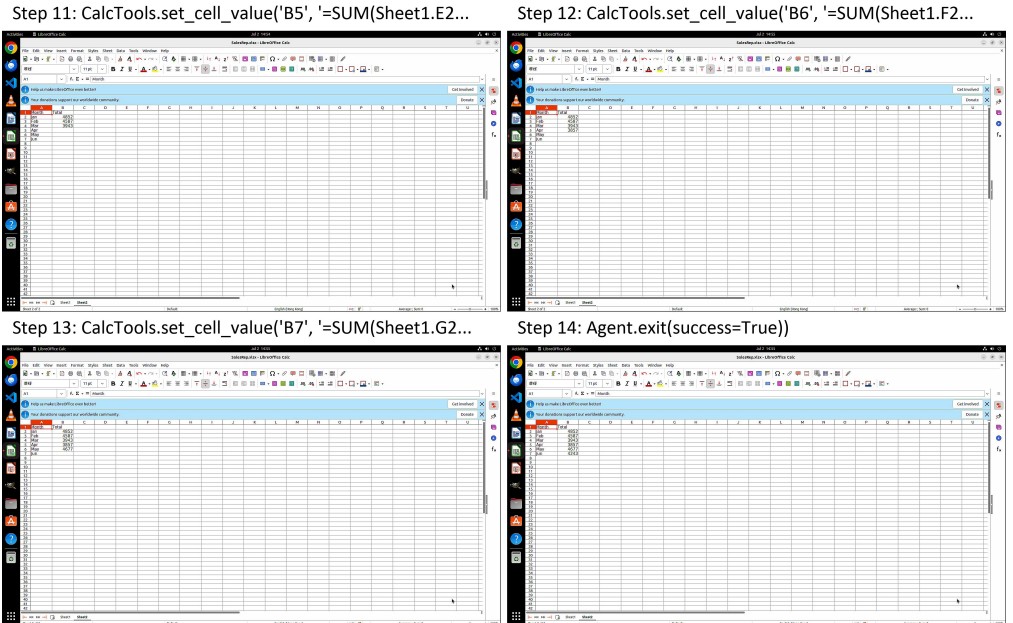

Figure 12: Task (Step 11-14): Create a table with two headers ("Month" and "Total") in a new sheet to show the total sales for all months.

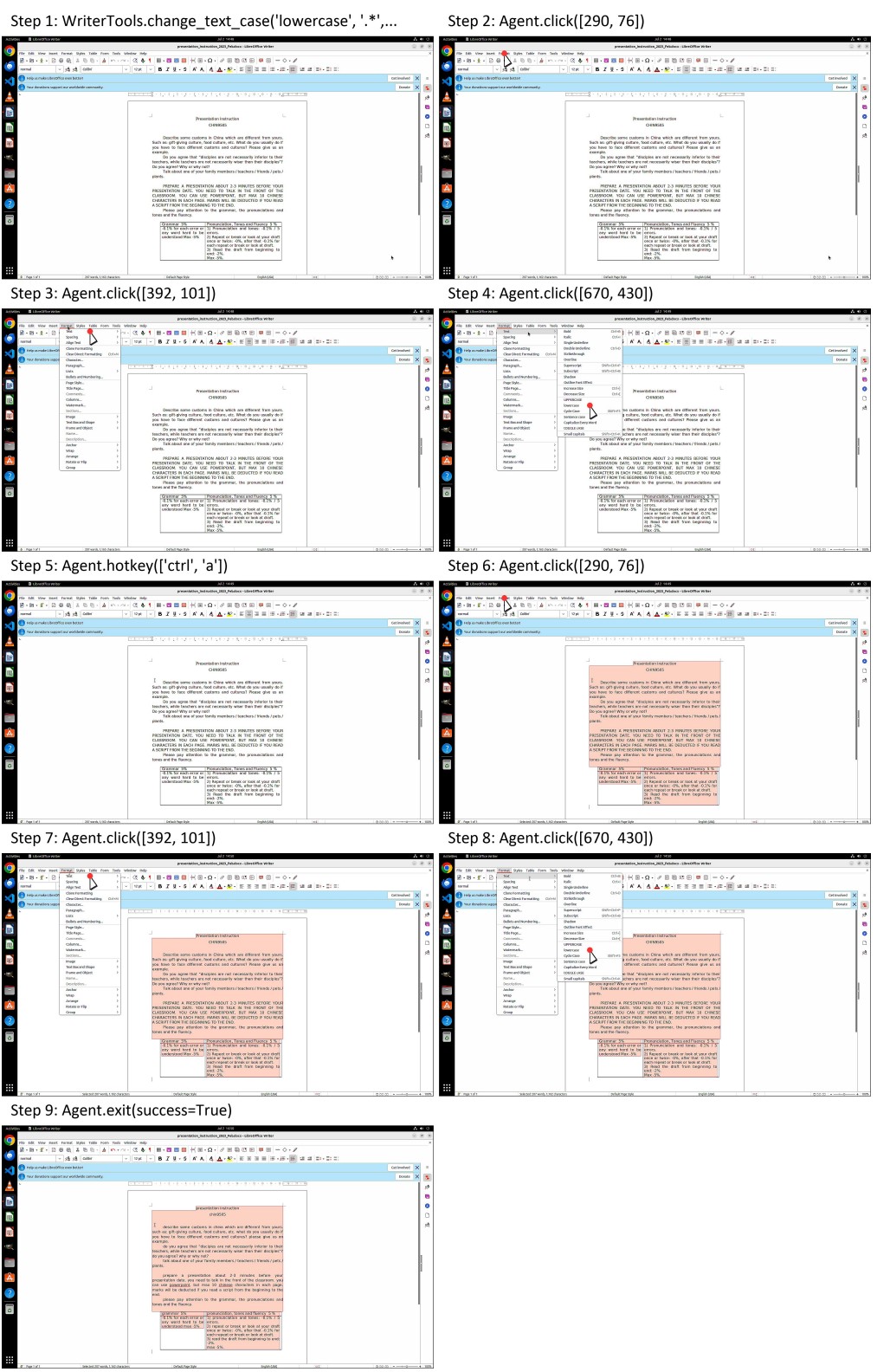

Figure 13: Task: I am currently engaged in text processing and require assistance in converting all uppercase text to lowercase within my document. This precision is critical for maintaining a uniform and polished presentation. Could you help me on this?

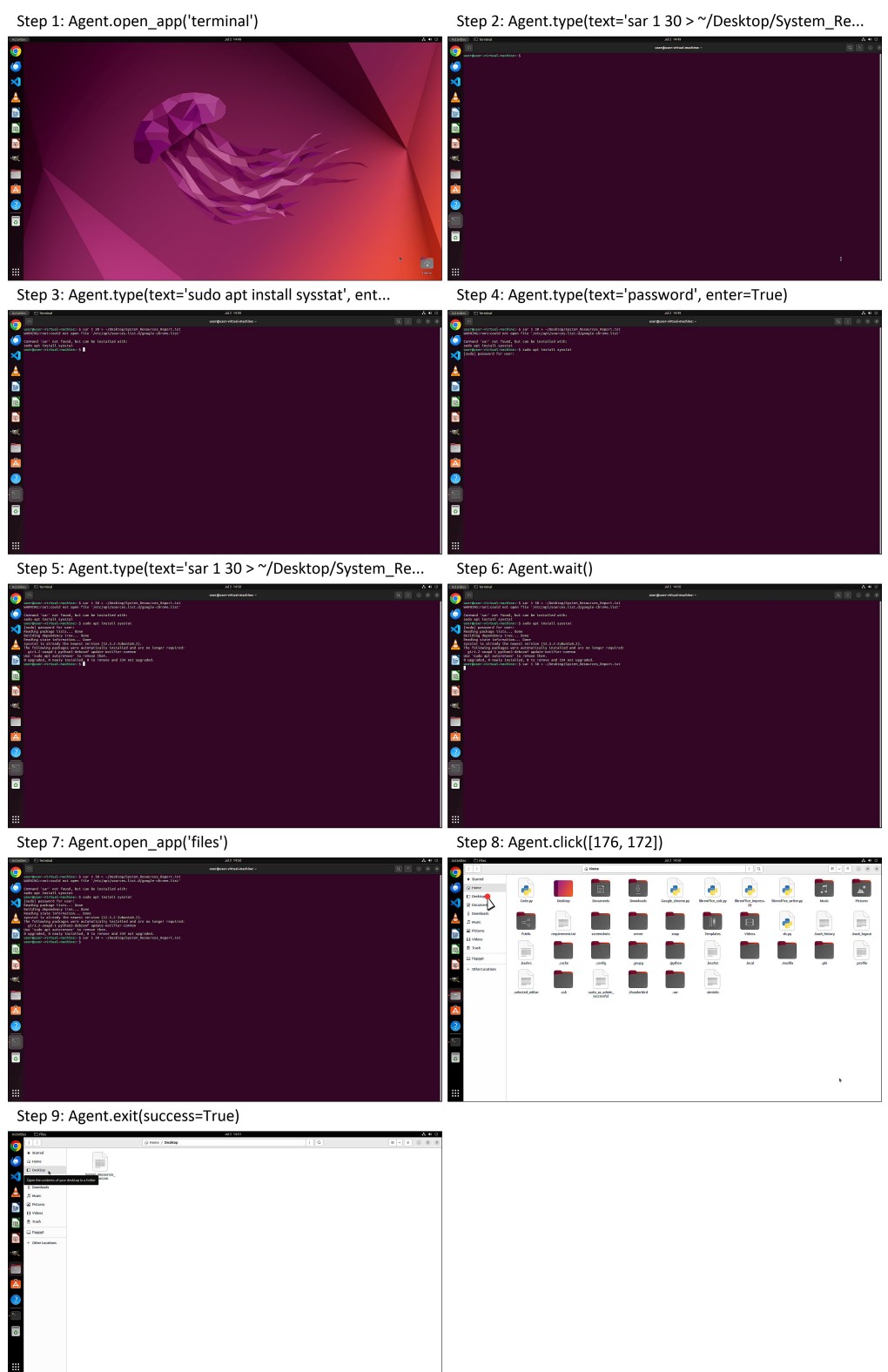

Figure 14: Task: Please use the 'sar' command in the 'sysstat' toolkit to monitor system activity, evaluate the status once every second for 30 seconds, output the results to "System_Resources_Report.txt" under Desktop.

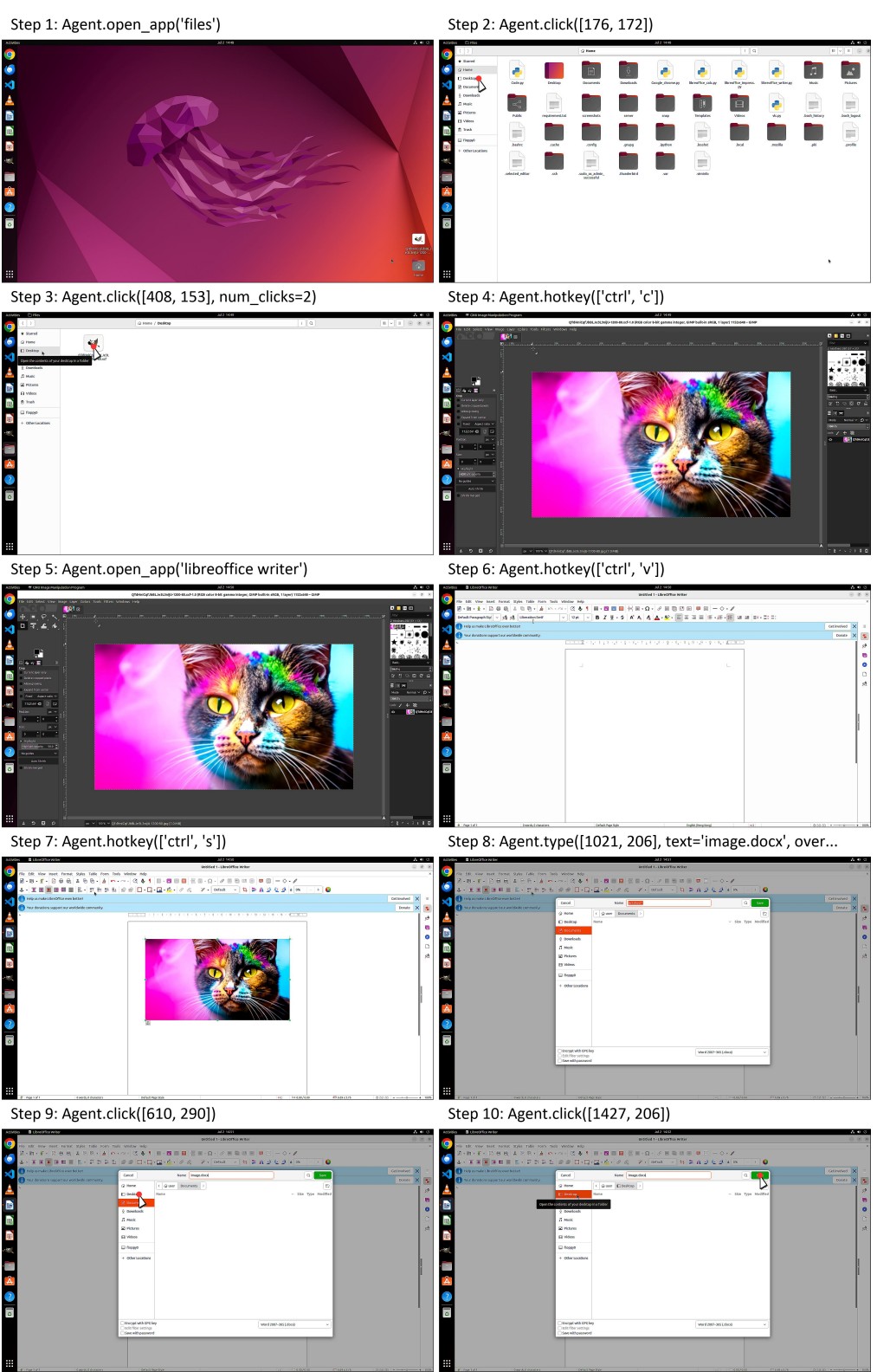

Figure 15: Task: I've stored my .xcf file on the Desktop. Can you assist me in copying the image and pasting it into a LibreOffice Writer document? Save the document as 'image.docx' on the Desktop, please.

Step 1: Agent.type(text="echo -e '1\2\3' | sed 's/$/<br...    Step 2: Agent.exit(success=True)

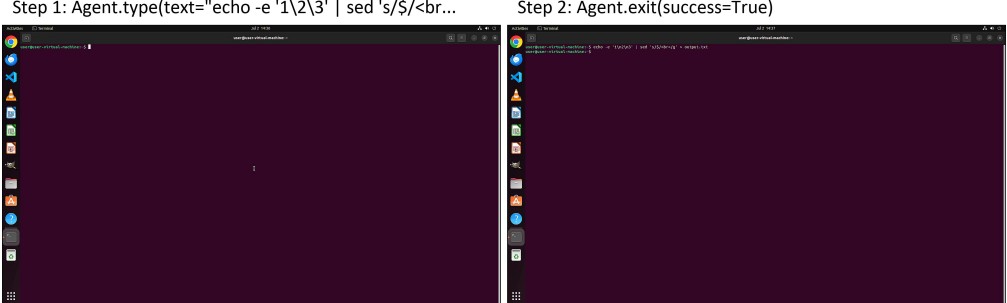

Figure 16: Fail Task (Question Misunderstanding Error): Append "
" to the end of each line in "1\n2\n3" and save in output.txt

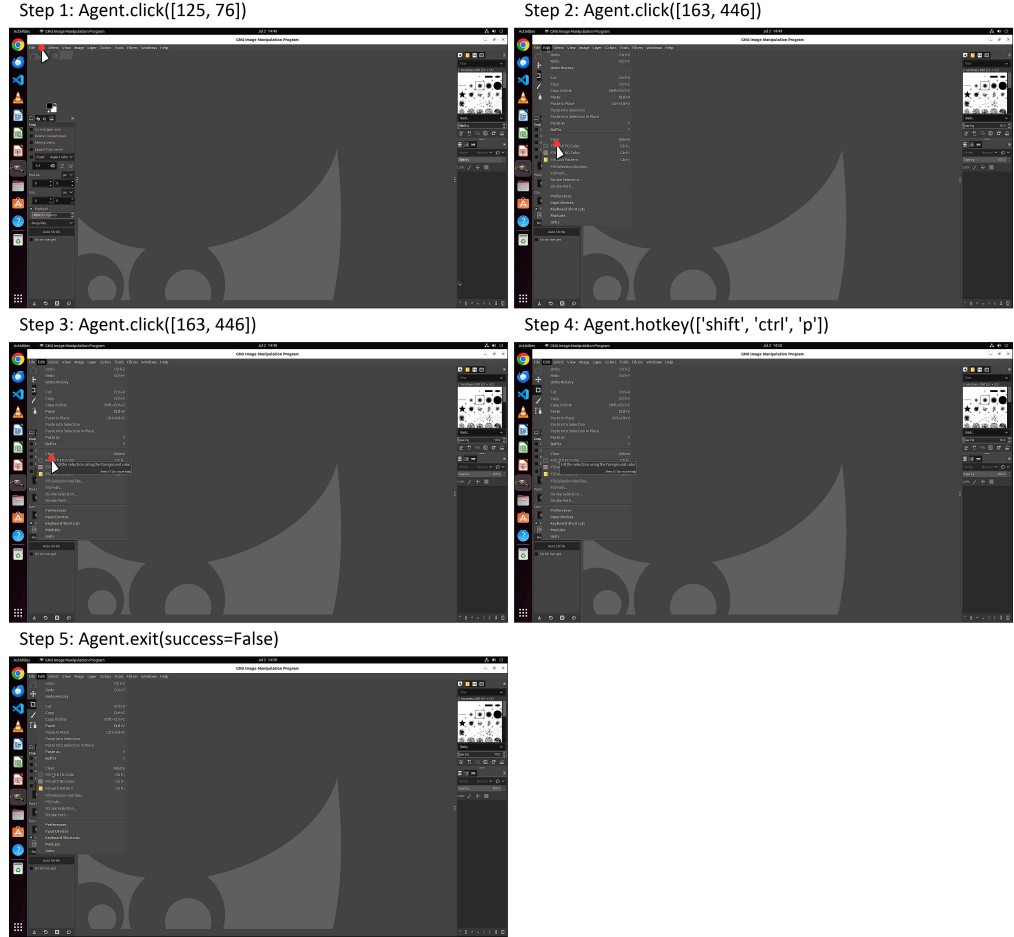

Figure 17: Fail Task (Click Operation Error): Please help change GIMP's theme from dark to light.

# I   THE USE OF LARGE LANGUAGE MODELS

During the preparation of this work, we employed LLMs to assist with language refinement and grammar correction. All research ideas, methodologies, experiments, and analyses were independently conceived, designed, and validated by the authors.

