# OpenReview forum: "ComputerRL: Scaling End-to-End Online Reinforcement Learning for Computer Use Agents"
_ICLR.cc/2026/Conference — ICLR 2026 Poster_

### Official Review · Reviewer_fU96 · 2025-10-29

**Soundness:** 2
**Presentation:** 2
**Contribution:** 2
**Rating:** 4
**Confidence:** 4

**Summary:**

This paper introduces ComputerRL, a large-scale framework for training computer use agents that can operate desktop environments via a unified API-GUI interaction paradigm. The framework integrates (1) a distributed RL infrastructure supporting thousands of parallel virtual desktops, (2) an automated pipeline for generating APIs from LLMs to augment GUI control, and (3) a training strategy named Entropulse, which alternates between reinforcement learning (RL) and supervised fine-tuning (SFT) to mitigate entropy collapse during long training runs.
The proposed system is evaluated on the OSWorld and OfficeWorld benchmarks, achieving state-of-the-art results (48.9% success rate) on computer automation tasks with open-weight models.

**Strengths:**

The API-GUI unification is an interesting engineering contribution that bridges the gap between human-designed interfaces and agent-level programmatic control.

The distributed RL infrastructure is impressive in scale and demonstrates strong engineering capability, enabling parallelized desktop environments at large scale.

The Entropulse idea addresses an important issue in long-horizon RL (entropy collapse), and the empirical results suggest measurable benefits in maintaining exploration and training stability.

The evaluation across multiple benchmarks (OSWorld, OfficeWorld) provides strong empirical evidence of performance improvements.

**Weaknesses:**

The paper’s novelty lies primarily in implementation and scaling, not in new algorithmic contributions. The API-GUI paradigm is conceptually straightforward—it effectively automates API construction via LLMs rather than introducing a new interaction or reasoning mechanism. Similarly, the Entropulse training alternation between RL and SFT is more of a practical training schedule than a novel learning algorithm.

The training curves in the figures appear to correspond to single runs, with no error bars or indication of variance across seeds. It is therefore unclear how stable the reported improvements are. The lack of information about the number of runs or random seeds undermines the reliability of the reported trends.

The paper does not include prior literature on diversity or exploration in RL, which would be relevant given the focus on entropy restoration and SFT alternation. Existing works in exploration-based RL, policy diversity, or ensemble-based approaches could provide valuable context, but they are not mentioned.

Given the large-scale infrastructure described (thousands of virtual desktops and 9B-parameter models), the paper lacks concrete information on hardware setup, training cost, and total compute used. For a work that emphasizes scalability, this omission is significant. Details such as training duration, GPU/CPU utilization, and cluster size should be clearly reported to assess feasibility and reproducibility.

**Questions:**

Could you clarify what you consider the core algorithmic innovation of ComputerRL beyond the engineering scale-up?

You mention that Entropulse increases exploration and behavioral diversity — could you provide quantitative evidence (e.g., entropy statistics, trajectory variance, or action coverage) supporting this claim?

Could you provide details on the hardware setup used — such as GPU type/count, CPU cluster size, memory, and network bandwidth?

How does your approach compare to prior work addressing entropy collapse or exploration in RL (e.g., maximum entropy RL, population-based methods)?

---

> ### Author Response · Authors · 2025-11-20
> **Response to Reviewer fU96 (1/4)**
>
> We sincerely thank you for the careful assessment and highly constructive feedback. We are grateful for your recognition of our work’s key strengths: the distributed RL infrastructure that scales to thousands of desktops, the unified API–GUI paradigm that improves efficiency and generalization for desktop automation, and the Entropulse strategy that effectively mitigates entropy collapse in long-horizon RL. Below, we respond to each of your comments point by point.
>
> ## Question about algorithmic novelty:
> We appreciate your feedback and would like to clarify that the core contribution of ComputerRL is not merely a theoretical innovation at the algorithmic level, but rather the systematic resolution of **practical challenges intrinsic to large-scale RL training for LLMs**.
> Conventional RL studies have predominantly been conducted under small-scale, low-compute settings. However, **large-scale RL training for computer control tasks presents fundamentally different challenges** that have not been thoroughly explored.
>
> 1. Environment Scaling: Overcoming Infrastructure Bottlenecks for Agentic RL
> While scaling for pre-training and SFT has been well studied, **scaling RL remains a longstanding difficulty**. Only recently have works such as DeepSeek-R1 demonstrated partial success in scaling RL for reasoning tasks. Agent-based RL faces even greater complexity when interacting with real, heterogeneous environments—especially real computer operating systems—rather than simulated domains.
> For computer control tasks, stable environment scaling is a prerequisite for successful agentic RL. In contrast to simulated environments, real OS environments exhibit high heterogeneity, unpredictability, and complex state spaces: virtual machines may crash, applications may become unresponsive, and network latency may fluctuate. When scaling to hundreds or even thousands of concurrent instances, such issues amplify exponentially.
> Our work is among the **first systematic studies to achieve scalable, stable RL training across more than 1,000 concurrent Ubuntu OS instances**, sustained for several days. Our asynchronous experience collection architecture, fault-tolerance mechanisms, and adaptive environment management strategies enable this capability. We regard the development of this stable, scalable environment infrastructure as a **substantial engineering contribution**, foundational for subsequent agentic RL research and a necessary condition for effective training.
>
> 2. API–GUI Paradigm: Addressing the Challenges of Training Efficiency and Stability
> Even with robust environment scaling, purely GUI-based RL training faces severe efficiency and stability constraints. GUI task execution often requires long action sequences (e.g., 5–10 clicks and keystrokes for a basic file manipulation), limiting rollout throughput and introducing deeper difficulties: **the semantic granularity of individual GUI actions is significantly less precise than that of API calls**, complicating reward assignment and destabilizing training.
> We propose an **API–GUI hybrid interaction paradigm** in which high-level task objectives are solved with mixed sequences of API calls and GUI operations. This reduces the required steps by approximately **1/3**, dramatically accelerating rollouts and enabling feasible large-scale parallel training. More importantly, semantic density per step increases significantly, yielding clearer supervision signals: API calls provide explicit intermediate goals and verifiable state transitions, stabilizing training dynamics and mitigating the exploration inefficiency and sparse rewards typical of purely GUI-based control.
>
> 3. Entropulse: Mitigating Entropy Collapse in Long-Horizon, Large-Scale Training
> Most existing agentic RL work trains under relatively small conditions (e.g., ~100 update steps), where the natural decline of policy entropy rarely forms a bottleneck. At scale—**thousands of concurrent environments, tens of hours of training, millions of agent–environment interactions**—we observed a phenomenon largely absent in smaller experiments yet fatal at scale: entropy collapse leading to premature convergence to suboptimal policies, stagnated exploration, and halted performance improvement. The necessity and design of Entropulse follow these observations and constraints:
> - At ~180 update steps in large-scale runs, policy entropy falls below a critical threshold, sharply reducing exploration and flattening learning curves.
> - Recently proposed entropy managing approaches (e.g., DAPO), while mitigating the entropy collapse problem, introduce significant overhead or affect convergence speed.
> - Entropulse cyclically refreshes the policy via SFT, restoring policy entropy effectively at low cost without degrading task performance.

---

> ### Author Response · Authors · 2025-11-20
> **Response to Reviewer fU96 (2/4)**
>
> This technique represents an **engineering-oriented innovation**: it is not a novel RL algorithm in the traditional theoretical sense, but a pragmatic solution grounded in problem observation, constraints analysis, and empirical validation. Our ablation studies show that removing Entropulse results in ~11% performance degradation in challenging workflow tasks.
>
> ### Contribution Positioning
> The value of ComputerRL lies **not** in proposing a fundamentally new RL theoretical algorithm, but in:
> 1. **Identifying and quantifying critical bottlenecks** in large-scale RL training (environment scaling, training stability and efficiency, entropy collapse);
> 2. **Providing practical, validated solutions** (scalable parallel infrastructure, API–GUI paradigm, Entropulse);
> 3. **Establishing reproducible benchmarks** via open-source code and detailed engineering documentation, enabling the community to extend upon our work.
> This type of engineering-driven innovation holds irreplaceable value in AI research. Analogous to how GPT-3’s pivotal contribution was not altering the Transformer architecture but **demonstrating emergent capabilities via scaling**, ComputerRL offers the **systematic demonstration of extending RL to real-world computer control tasks with practical-level performance**.
>
> ## Missing Multi-Run Evaluations and Error Bars in Training Curves:
> Thank you for raising concerns about reproducibility and statistical rigor in our results. We would like to clarify that in our original manuscript, we have already included confidence intervals in our training curves and provided detailed information about random seeds and training hyperparameters in Appendix D.
> To further strengthen the reliability of our method, we include experimental results from multiple runs with different base models. In the revised manuscript, we now report:
> - Comprehensive statistical indicators in Appendix E (Additional Experimental Indicators)
> - Average reward and confidence intervals across repetitive experimental runs in Appendix F (Repetitive Experiments with Different Base Models)
>
> These updates provide stronger empirical support for the stability and reproducibility of our proposed method. We believe these additions adequately address concerns about multi-run experiments and the statistical significance of our reported results.
>
> ```latex
> \section{Additional Experimental Indicators}\label{apdx:exp_indicators}
> To more comprehensively validate the effectiveness of our method, we report detailed experimental indicators to support our conclusions, as shown in Figure~\ref{figs:exp_indicators}. These indicators include Average Reward, Entropy Loss, KL Loss, PPO KL, Average Margin, BoN Reward, Average Turns, and Response Length. Based on these metrics, we make the following observations:
>
> \begin{itemize}
>     \item Entropulse effectively increases the stochasticity of the policy, leading to a substantial improvement in BoN after activation. This, in turn, drives the growth of the margin and enables the policy to continue learning and improving.
>     \item After applying Entropulse, the response patterns of the policy (including response length and number of dialogue turns) become closer to those before the first-stage RL training (i.e., shorter), while maintaining comparable scores. This indicates that Entropulse helps the policy discover better solutions along shorter trajectories, thereby suppressing excessive reasoning and redundant steps.
>     \item After resetting the reference model, the KL Loss is also reset, allowing the policy to explore a larger space relative to the new reference. This prevents the policy from being overly constrained by its previous strategy.
> \end{itemize}
>
> \section{Repetitive Experiments with Different Base Models}\label{apdx:reproducible_exp}
> To further verify the effectiveness of our method, we conduct repetitive experiments with different base models (both text and multimodal), demonstrating the stability and superiority of our approach. The results are reported in Figure~\ref{figs:repetitive_exp}.
> ```

---

> ### Author Response · Authors · 2025-11-20
> **Response to Reviewer fU96 (3/4)**
>
> ## Coverage of Literature and Comparison with Prior Entropy-Management Approaches:
> Thank you for highlighting this gap. In the revised Section 5, “RL and Entropy Management for LLMs,” we now cite core explorations in maximum-entropy RL, ensemble methods, and recent LLM entropy-collapse studies (e.g., DAPO and token-level interventions). We also add a paragraph comparing Entropulse to these works, emphasizing that our cyclic SFT–based entropy refresh sustains exploration during RL training while remaining both computationally efficient and conceptually simple.
>
> ```latex
> \vpara{RL and Entropy Management for LLMs.}
> PPO~\citep{schulman2017proximal} addresses instability in policy gradients for RL training. GRPO~\citep{guo2025deepseek} extends PPO with group sampling and removes value updates. Maximum entropy RL~\citep{haarnoja2018soft} and ensemble methods~\citep{lee2021sunrise, de2025enhancing} maintain diversity through regularization or multiple models. Recent work identifies entropy collapse as a critical challenge in LLM RL~\citep{cui2025entropy}, with proposed solutions including DAPO~\citep{yu2025dapo} with adaptive clipping and token-level interventions~\citep{hao2025rethinking}. Entropulse takes a different approach by actively restoring collapsed entropy through targeted SFT training on diverse rollout data, achieving extended training.
> ```

---

> ### Author Response · Authors · 2025-11-20
> **Response to Reviewer fU96 (4/4)**
>
> ## Experimental Setup for Reproducibility and Feasibility Assessment:
> We appreciate your feedback. To facilitate reproducible research, Appendix D of the original manuscript already contained the complete set of training parameters, including the cluster scale (16 nodes) and SFT and RL hyperparameters (e.g., seed numbers). We also provide our source code at https://anonymous.4open.science/r/ComputerRL-ICLR-758E/. To further support replication and potential methodological enhancements, we provide comprehensive information on the computing infrastructure, training duration, and estimated computational cost below. These updates are synchronized with Appendix D in the revised manuscript.
>
> ---
>
> ### **Hardware Configuration**
>
> #### **Training Cluster Overview**
> | Parameter | Specification |
> |-----------|---------------|
> | Cluster size | 16 nodes |
> | GPU type | NVIDIA H800 |
> | GPUs per node | 8 |
> | Total GPUs | 128 (16 × 8) |
>
> #### **CPU Configuration**
> | Parameter | Specification |
> |-----------|---------------|
> | Model | Intel Xeon Gold 6430 |
> | Architecture | x86_64 |
> | Sockets | 2 |
> | Cores per socket | 32 |
> | Total cores | 64 |
> | Threads | 64 (1 thread per core) |
> | Base frequency | 2.1 GHz |
> | Minimum frequency | 800 MHz |
> | Instruction set extensions | AVX-512, AVX512_FP16, AMX (INT8/BF16/Tile) |
>
> #### **Cache Hierarchy**
> | Level | Capacity |
> |-------|----------|
> | L1 Data Cache | 3 MiB (64 instances) |
> | L1 Instruction Cache | 2 MiB (64 instances) |
> | L2 Cache | 128 MiB (64 instances) |
> | L3 Cache | 120 MiB (2 instances) |
>
> #### **Memory Configuration**
> | Parameter | Specification |
> |-----------|---------------|
> | Total capacity | 2.0 TiB |
> | Available memory | 1.9 TiB |
> | NUMA nodes | 2 |
> | NUMA Node 0 CPUs | 0–31 |
> | NUMA Node 1 CPUs | 32–63 |
> | Swap | Disabled (0 B) |
>
> #### **Networking**
> | Parameter | Specification |
> |-----------|---------------|
> | Interconnect | InfiniBand / high-speed Ethernet |
> | Address width | Physical 46-bit, Virtual 57-bit |
>
> ---
>
> ### **Environment Cluster for Distributed RL**
> A dedicated environment simulation cluster was used for parallelized RL environment execution:
>
> | Parameter | Specification |
> |-----------|---------------|
> | Cluster size | 7 nodes |
>
> #### **CPU Configuration**
> | Parameter | Specification |
> |-----------|---------------|
> | Model | Intel Xeon 6986P-C (Granite Rapids) |
> | Architecture | x86_64 |
> | Sockets | 1 |
> | Cores per socket | 120 |
> | Total cores | 120 |
> | Threads | 240 (2 threads per core, hyper-threaded) |
> | Base frequency | 3.3 GHz |
> | Max turbo frequency | 3.9 GHz |
> | Minimum frequency | 800 MHz |
> | Instruction set extensions | AVX-512, AVX512_BF16, AMX, SHA-NI |
>
> #### **Cache Hierarchy**
> | Level | Capacity |
> |-------|----------|
> | L1 Data Cache | 5.6 MiB |
> | L1 Instruction Cache | 7.5 MiB |
> | L2 Cache | 240 MiB |
> | L3 Cache | 504 MiB |
>
> #### **Memory Configuration**
> | Parameter | Specification |
> |-----------|---------------|
> | Memory per node | 1.1 TiB |
> | Available memory | 949 GiB |
> | NUMA nodes | 3 |
> | NUMA Node 0 CPUs | 0–39, 120–159 |
> | NUMA Node 1 CPUs | 40–79, 160–199 |
> | NUMA Node 2 CPUs | 80–119, 200–239 |
> | Swap | Disabled (0 B) |
> | Total cluster memory | ~7.7 TiB (7 nodes) |
>
> #### **Virtualization and Features**
> | Parameter | Specification |
> |-----------|---------------|
> | Virtualization | Intel VT-x, EPT, VPID |
> | Security mitigations | Enhanced IBRS, IBPB, full Spectre/Meltdown mitigation |
> | Cryptographic acceleration | AES-NI, SHA-NI, AVX512_VAES |
> | AI acceleration | AMX, AVX512_BF16, AVX512_VNNI |
> | Address width | Physical 52-bit, Virtual 57-bit |
>
> ---
>
> ### **Virtual Environment Instances for RL**
> Each RL environment ran in a lightweight virtual machine:
>
> | Parameter | Specification |
> |-----------|---------------|
> | OS image | Ubuntu 20.04 LTS |
> | vCPU cores | 2 |
> | Memory allocation | 4 GB |
> | Mean runtime network bandwidth | 0.4 Mbps |
> | Virtualization platform | KVM/QEMU |
>
> *Note:* Each instance provided an independent Ubuntu 20.04 desktop environment for GUI-based RL tasks. The minimal configuration (2 cores, 4 GB RAM) was chosen to maximize parallelism under constrained hardware resources, enabling large-scale distributed RL training.
>
> ---
>
> ### **Training Duration and FLOP Estimates (Multimodal Example)**
>
> **SFT (Behavior Cloning)**
> - Duration: 16 hours
> - Total FLOPs: 1.67 × 10^16
>
> **SFT (Entropulse)**
> - Duration: 11 hours
> - Total FLOPs: 1.22 × 10^16
>
> **Two-Stage RL**
> - Duration: 58 hours
> - Total FLOPs (estimated): 3.21 × 10^17

---

> ### Author Response · Authors · 2025-11-27
>
> Dear Reviewer,
>
> I hope this message finds you well. As the discussion period is approaching its end and less than one week remains, we would like to ensure that all of your questions and concerns regarding our submission (*ComputerRL*) have been fully addressed.
>
> If there are any additional points, clarifications, or feedback you would like us to consider, please feel free to share them at your earliest convenience. Your insights are highly valuable to us, and we are eager to address any remaining issues to improve the work and facilitate the review process.
>
> Thank you very much for your time and effort in evaluating our paper. We greatly appreciate your contributions and look forward to any further comments you may have.
>
> Best regards,
> The Authors of *ComputerRL*

---

> > ### Comment · Reviewer_fU96 · 2025-11-27
> > **Final Score**
> >
> > Thank you for the detailed and comprehensive responses. I appreciate the clarifications regarding the engineering contributions, the entropy-management motivation, the expanded experimental statistics, and the additional details on compute and reproducibility. While these revisions address several of my earlier concerns, they do not substantially change my overall assessment of the paper’s novelty and significance.
> > In particular, issues related to entropy-driven exploration and the role of parallelism remain central. I agree that large-scale RL training for computer control tasks presents unique challenges, but I would expect this to motivate the development of new algorithmic insights or methodological innovations,beyond engineering scale,up alone,that could help form the basis of future literature on efficient parallel RL for complex environments.
> >
> > Accordingly, I will maintain my original score

---

### Official Review · Reviewer_oLgc · 2025-10-31

**Soundness:** 3
**Presentation:** 4
**Contribution:** 3
**Rating:** 6
**Confidence:** 3

**Summary:**

This paper presents **ComputerRL**, a framework for training computer use agents capable of performing desktop operations through a unified API-GUI interaction paradigm. The framework integrates:
1. a distributed RL infrastructure that supports training across thousands of virtual desktop instances;
2. aan LLM-driven module that automatically derives application programming interfaces (APIs) to extend and complement traditional GUI-based controls;
3. Entropulse, a hybrid training regime that periodically alternates between RL and SFT phases to counteract entropy collapse and maintain exploration over extended training trajectories.

Evaluations on OSWorld and OfficeWorld benchmarks show state-of-the-art results (48.9% success rate) using open-weight models (GLM-4-9B-0414, GLM-4.1V-9B-Thinking), outperforming both proprietary and open baselines such as OpenAI CUA, Claude 4.0, and UI-TARS.

**Strengths:**

- **Strong systems and engineering contribution:** The distributed RL infrastructure is technically impressive, enabling large-scale online RL across thousands of virtualized desktop environments. Such scale is rare in open research and represents a substantial engineering achievement.

- **Practical API-GUI paradigm:**  The unified action space combining GUI operations with automatically constructed APIs addresses a key bottleneck in desktop automation. The LLM-driven API construction pipeline is pragmatic and lowers the barrier for generalization.

- **Entropulse training strategy:** Alternating RL and SFT phases effectively combats entropy collapse and stabilizes long-horizon training. While conceptually simple, it appears empirically effective and easy to adopt in practice.

- **Empirical performance:** The results on both OSWorld and OfficeWorld are strong and consistent. The proposed approach achieves superior performance and sample efficiency (fewer steps per task) compared to all evaluated baselines. Ablation studies indicate the importance of multi-stage training and the API-GUI design.

- **Writing and organization:** The paper is very well written, clearly structured, and visually well presented.

**Weaknesses:**

- **Unsubstantiated claims about diversity and exploration:** The paper claims that alternating SFT with RL increases exploration and diversity, yet no quantitative evidence is provided. Metrics such as action entropy, trajectory variance, or coverage are not analyzed. The only evidence is a qualitative entropy curve, which is insufficient.

- **Incomplete empirical rigor and reproducibility:** All training curves appear to represent single runs without confidence intervals or variance estimates. The number of seeds, randomization strategy, or statistical robustness is not discussed. Given the scale (9B-parameter models, thousands of desktops), compute and reproducibility details are critically missing — no information on hardware setup, training duration, or cluster configuration. For a paper emphasizing scalability, this omission is significant.

- **Limited methodological novelty:** The paper’s main innovations are engineering-oriented. The API-GUI paradigm is conceptually straightforward since it. The novelty lies in the automation pipeline, not the interaction paradigm itself. Similarly, Entropulse is a *training schedule* rather than a new RL algorithm; no theoretical or comparative justification is provided beyond empirical observations.

- **Missing broader impact and ethical discussion:** Given that this system trains autonomous agents capable of full computer control, a discussion of potential misuse, safety mechanisms, or privacy implications is notably absent.

**Questions:**

- Please specify the number of seeds, compute hardware, and total training cost (GPU hours, cluster size). How reproducible are the reported results on smaller scales?
- What are the hardware requirements and cost implications for scaling your distributed RL infrastructure to the reported scale? What performance trade-offs or bottlenecks did you observe in practice?

**Details Of Ethics Concerns:**

I just want to pinpoint that there is a human annotation protocol that is carefully explained by authors in appendix E.

---

> ### Author Response · Authors · 2025-11-20
> **Response to Reviewer oLgc (1/5)**
>
> Thank you for carefully reading our paper and for the encouraging evaluation. We appreciate your recognition of our large-scale distributed RL infrastructure across thousands of desktops as a strong systems and engineering contribution, the unified API–GUI paradigm and LLM-driven API construction pipeline for generalizable desktop automation, and the effectiveness and ease of adoption of the Entropulse training strategy in mitigating entropy collapse. Below, we respond to each of your questions in detail.
>
> ## Unsubstantiated claims about diversity and exploration:
> Thank you for this valuable comment. Our discussion of our experiments in the original manuscript may not have been sufficiently detailed. To address this, we have added a more comprehensive set of experimental indicators in Appendix E (Additional Experimental Indicators) of the revised manuscript to better substantiate our claims, as summarized below.
>
> ```latex
> To more comprehensively validate the effectiveness of our method, we report detailed experimental indicators to support our conclusions, as shown in Figure~\ref{figs:exp_indicators}. These indicators include Average Reward, Entropy Loss, KL Loss, PPO KL, Average Margin, BoN Reward, Average Turns, and Response Length. Based on these metrics, we make the following observations:
>
> \begin{itemize}
>     \item Entropulse effectively increases the stochasticity of the policy, leading to a substantial improvement in BoN after activation. This, in turn, drives the growth of the margin and enables the policy to continue learning and improving.
>     \item After applying Entropulse, the response patterns of the policy (including response length and number of dialogue turns) become closer to those before the first-stage RL training (i.e., shorter), while maintaining comparable scores. This indicates that Entropulse helps the policy discover better solutions along shorter trajectories, thereby suppressing excessive reasoning and redundant steps.
>     \item After resetting the reference model, the KL Loss is also reset, allowing the policy to explore a larger space relative to the new reference. This prevents the policy from being overly constrained by its previous strategy.
> \end{itemize}
> ```
>
> ## Incomplete empirical rigor and reproducibility:
> Thank you for raising concerns about reproducibility and statistical rigor in our results. We want to clarify that in our original manuscript, we have already included confidence intervals in our training curves and provided detailed information about random seeds and training hyperparameters in Appendix D. We also provide our source code at https://anonymous.4open.science/r/ComputerRL-ICLR-758E/.
>
> To further strengthen the reliability of our method, we additionally conduct repetitive experiments with different base models and report the corresponding statistical results. In the revised manuscript, we add average rewards and confidence intervals across repetitive experimental runs with different base models in Appendix F (Repetitive Experiments with Different Base Models), demonstrating the stability and robustness of our method. We believe these additions can address concerns about multi-run experiments and the statistical significance of our reported results.
>
> ```latex
> \section{Repetitive Experiments with Different Base Models}\label{apdx:reproducible_exp}
> To further verify the effectiveness of our method, we conduct repetitive experiments with different base models (both text and multimodal), demonstrating the stability and superiority of our approach. The results are reported in Figure~\ref{figs:repetitive_exp}.
> ```
>
> To further support replication and potential methodological enhancements, we provide comprehensive information on the computing infrastructure, training duration, and estimated computational cost below. These updates are synchronized with Appendix D in the revised manuscript.

---

> ### Author Response · Authors · 2025-11-20
> **Response to Reviewer oLgc (2/5)**
>
> ### **Hardware Configuration**
>
> #### **Training Cluster Overview**
> | Parameter | Specification |
> |-----------|---------------|
> | Cluster size | 16 nodes |
> | GPU type | NVIDIA H800 |
> | GPUs per node | 8 |
> | Total GPUs | 128 (16 × 8) |
>
> #### **CPU Configuration**
> | Parameter | Specification |
> |-----------|---------------|
> | Model | Intel Xeon Gold 6430 |
> | Architecture | x86_64 |
> | Sockets | 2 |
> | Cores per socket | 32 |
> | Total cores | 64 |
> | Threads | 64 (1 thread per core) |
> | Base frequency | 2.1 GHz |
> | Minimum frequency | 800 MHz |
> | Instruction set extensions | AVX-512, AVX512_FP16, AMX (INT8/BF16/Tile) |
>
> #### **Cache Hierarchy**
> | Level | Capacity |
> |-------|----------|
> | L1 Data Cache | 3 MiB (64 instances) |
> | L1 Instruction Cache | 2 MiB (64 instances) |
> | L2 Cache | 128 MiB (64 instances) |
> | L3 Cache | 120 MiB (2 instances) |
>
> #### **Memory Configuration**
> | Parameter | Specification |
> |-----------|---------------|
> | Total capacity | 2.0 TiB |
> | Available memory | 1.9 TiB |
> | NUMA nodes | 2 |
> | NUMA Node 0 CPUs | 0–31 |
> | NUMA Node 1 CPUs | 32–63 |
> | Swap | Disabled (0 B) |
>
> #### **Networking**
> | Parameter | Specification |
> |-----------|---------------|
> | Interconnect | InfiniBand / high-speed Ethernet |
> | Address width | Physical 46-bit, Virtual 57-bit |
>
> ---
>
> ### **Environment Cluster for Distributed RL**
> A dedicated environment simulation cluster was used for parallelized RL environment execution:
>
> | Parameter | Specification |
> |-----------|---------------|
> | Cluster size | 7 nodes |
>
> #### **CPU Configuration**
> | Parameter | Specification |
> |-----------|---------------|
> | Model | Intel Xeon 6986P-C (Granite Rapids) |
> | Architecture | x86_64 |
> | Sockets | 1 |
> | Cores per socket | 120 |
> | Total cores | 120 |
> | Threads | 240 (2 threads per core, hyper-threaded) |
> | Base frequency | 3.3 GHz |
> | Max turbo frequency | 3.9 GHz |
> | Minimum frequency | 800 MHz |
> | Instruction set extensions | AVX-512, AVX512_BF16, AMX, SHA-NI |
>
> #### **Cache Hierarchy**
> | Level | Capacity |
> |-------|----------|
> | L1 Data Cache | 5.6 MiB |
> | L1 Instruction Cache | 7.5 MiB |
> | L2 Cache | 240 MiB |
> | L3 Cache | 504 MiB |
>
> #### **Memory Configuration**
> | Parameter | Specification |
> |-----------|---------------|
> | Memory per node | 1.1 TiB |
> | Available memory | 949 GiB |
> | NUMA nodes | 3 |
> | NUMA Node 0 CPUs | 0–39, 120–159 |
> | NUMA Node 1 CPUs | 40–79, 160–199 |
> | NUMA Node 2 CPUs | 80–119, 200–239 |
> | Swap | Disabled (0 B) |
> | Total cluster memory | ~7.7 TiB (7 nodes) |
>
> #### **Virtualization and Features**
> | Parameter | Specification |
> |-----------|---------------|
> | Virtualization | Intel VT-x, EPT, VPID |
> | Security mitigations | Enhanced IBRS, IBPB, full Spectre/Meltdown mitigation |
> | Cryptographic acceleration | AES-NI, SHA-NI, AVX512_VAES |
> | AI acceleration | AMX, AVX512_BF16, AVX512_VNNI |
> | Address width | Physical 52-bit, Virtual 57-bit |
>
> ---
>
> ### **Virtual Environment Instances for RL**
> Each RL environment ran in a lightweight virtual machine:
>
> | Parameter | Specification |
> |-----------|---------------|
> | OS image | Ubuntu 20.04 LTS |
> | vCPU cores | 2 |
> | Memory allocation | 4 GB |
> | Mean runtime network bandwidth | 0.4 Mbps |
> | Virtualization platform | KVM/QEMU |
>
> *Note:* Each instance provided an independent Ubuntu 20.04 desktop environment for GUI-based RL tasks. The minimal configuration (2 cores, 4 GB RAM) was chosen to maximize parallelism under constrained hardware resources, enabling large-scale distributed RL training.
>
> ---
>
> ### **Training Duration and FLOP Estimates (Multimodal Example)**
>
> **SFT (Behavior Cloning)**
> - Duration: 16 hours
> - Total FLOPs: 1.67 × 10^16
>
> **SFT (Entropulse)**
> - Duration: 11 hours
> - Total FLOPs: 1.22 × 10^16
>
> **Two-Stage RL**
> - Duration: 58 hours
> - Total FLOPs (estimated): 3.21 × 10^17

---

> ### Author Response · Authors · 2025-11-20
> **Response to Reviewer oLgc (3/5)**
>
> ## Limited methodological novelty:
> We appreciate your feedback and would like to clarify that the core contribution of ComputerRL is not merely a theoretical innovation at the algorithmic level, but rather the systematic resolution of **practical challenges intrinsic to large-scale RL training for LLMs**.
> Conventional RL studies have predominantly been conducted under small-scale, low-compute settings. However, **large-scale RL training for computer control tasks presents fundamentally different challenges** that have not been thoroughly explored.
>
> 1. Environment Scaling: Overcoming Infrastructure Bottlenecks for Agentic RL
> While scaling for pre-training and SFT has been well studied, **scaling RL remains a longstanding difficulty**. Only recently have works such as DeepSeek-R1 demonstrated partial success in scaling RL for reasoning tasks. Agent-based RL faces even greater complexity when interacting with real, heterogeneous environments—especially real computer operating systems—rather than simulated domains.
> For computer control tasks, stable environment scaling is a prerequisite for successful agentic RL. In contrast to simulated environments, real OS environments exhibit high heterogeneity, unpredictability, and complex state spaces: virtual machines may crash, applications may become unresponsive, and network latency may fluctuate. When scaling to hundreds or even thousands of concurrent instances, such issues amplify exponentially.
> Our work is among the **first systematic studies to achieve scalable, stable RL training across more than 1,000 concurrent Ubuntu OS instances**, sustained for several days. Our asynchronous experience collection architecture, fault-tolerance mechanisms, and adaptive environment management strategies enable this capability. We regard the development of this stable, scalable environment infrastructure as a **substantial engineering contribution**, foundational for subsequent agentic RL research and a necessary condition for effective training.
>
> 2. API–GUI Paradigm: Addressing the Challenges of Training Efficiency and Stability
> Even with robust environment scaling, purely GUI-based RL training faces severe efficiency and stability constraints. GUI task execution often requires long action sequences (e.g., 5–10 clicks and keystrokes for a basic file manipulation), limiting rollout throughput and introducing deeper difficulties: **the semantic granularity of individual GUI actions is significantly less precise than that of API calls**, complicating reward assignment and destabilizing training.
> We propose an **API–GUI hybrid interaction paradigm** in which high-level task objectives are solved with mixed sequences of API calls and GUI operations. This reduces the required steps by approximately **1/3**, dramatically accelerating rollouts and enabling feasible large-scale parallel training. More importantly, semantic density per step increases significantly, yielding clearer supervision signals: API calls provide explicit intermediate goals and verifiable state transitions, stabilizing training dynamics and mitigating the exploration inefficiency and sparse rewards typical of purely GUI-based control.
>
> 3. Entropulse: Mitigating Entropy Collapse in Long-Horizon, Large-Scale Training
> Most existing agentic RL work trains under relatively small conditions (e.g., ~100 update steps), where the natural decline of policy entropy rarely forms a bottleneck. At scale—**thousands of concurrent environments, tens of hours of training, millions of agent–environment interactions**—we observed a phenomenon largely absent in smaller experiments yet fatal at scale: entropy collapse leading to premature convergence to suboptimal policies, stagnated exploration, and halted performance improvement. The necessity and design of Entropulse follow these observations and constraints:
> - At ~180 update steps in large-scale runs, policy entropy falls below a critical threshold, sharply reducing exploration and flattening learning curves.
> - Recently proposed entropy managing approaches (e.g., DAPO), while mitigating the entropy collapse problem, introduce significant overhead or affect convergence speed.
> - Entropulse cyclically refreshes the policy via SFT, restoring policy entropy effectively at low cost without degrading task performance.
>
> This technique represents an **engineering-oriented innovation**: it is not a novel RL algorithm in the traditional theoretical sense, but a pragmatic solution grounded in problem observation, constraints analysis, and empirical validation. Our ablation studies show that removing Entropulse results in ~11% performance degradation in challenging workflow tasks.

---

> ### Author Response · Authors · 2025-11-20
> **Response to Reviewer oLgc (4/5)**
>
> ### Contribution Positioning
> The value of ComputerRL lies **not** in proposing a fundamentally new RL theoretical algorithm, but in:
> 1. **Identifying and quantifying critical bottlenecks** in large-scale RL training (environment scaling, training stability and efficiency, entropy collapse);
> 2. **Providing practical, validated solutions** (scalable parallel infrastructure, API–GUI paradigm, Entropulse);
> 3. **Establishing reproducible benchmarks** via open-source code and detailed engineering documentation, enabling the community to extend upon our work.
> This type of engineering-driven innovation holds irreplaceable value in AI research. Analogous to how GPT-3’s pivotal contribution was not altering the Transformer architecture but **demonstrating emergent capabilities via scaling**, ComputerRL offers the **systematic demonstration of extending RL to real-world computer control tasks with practical-level performance**.
>
> ## Missing broader impact and ethical discussion:
> Thank you for raising this important concern. We would like to clarify that safety and ethical considerations are discussed in Appendix H.3 (Foundations for Safe and Aligned Autonomy) of our paper. In this section, we explicitly acknowledge that "autonomous control over desktop platforms raises profound questions about safety, trustworthiness, and user agency." We recognize that the margin for error narrows dramatically when agents are empowered to modify files, access sensitive data, or execute unbounded actions. To mitigate these risks, we outline our roadmap for safe deployment, which includes:
> - Granular permissioning frameworks to control what actions agents can perform
> - Robust pre-action validation to verify operations before execution
> - Multi-stage approval protocols for sensitive operations requiring human oversight
> Our ultimate goal is to establish safety standards and best practices that can serve as foundational infrastructure not only for ComputerRL but for the broader ecosystem of autonomous desktop agents.

---

> ### Author Response · Authors · 2025-11-20
> **Response to Reviewer oLgc (5/5)**
>
> ## Hardware, Cost, and Bottlenecks in Scaling Distributed RL to Thousand-Instance Desktop Environments:
> We appreciate your interest in the scalability and practical deployment aspects of our system. In extending our distributed RL infrastructure to the reported scale, we have accumulated substantial hands-on experience concerning hardware requirements, cost implications, and performance trade-offs. We have updated these experiences in Appendix D in the revised manuscript.
>
> ### Hardware Requirements and System Configuration (Appendix D.2-5)
>
> **Minimum Running Configuration**
> - **Training Cluster** – Distributed RL training requires at least **4 GPU nodes** to run our training pipeline.
>
> **Resource Ratio Experience**
> In our empirically validated deployment:
> - Each GPU achieves optimal utilization when paired with approximately **80 rollouts**.
> - Each environment server can reliably host **200 concurrently running virtual environments**.
> This ratio maintains equilibrium between GPU computation and environment sampling, minimizing idle computational resources.
>
> **Cost Estimation**
> - Comprehensive hardware specifications—including CPU type, memory capacity, and network configuration—are documented in Appendix D.3 and D.4.
> - The aggregate GPU-hours and associated costs across the entire training cycle are detailed in Appendix D.5.
>
> ### Key Performance Trade-offs and Bottlenecks (Appendix D.6)
>
> Our observations identify two principal hyperparameters that significantly influence the balance between training efficiency and convergence stability:
>
> **1. Responses per Prompt (RpP) – Exploration Upper Bound vs. Sampling Efficiency**
> - **Role** – RpP defines the breadth of the search space in Best-of-N (BoN) sampling.
> - **Trade-off**:
>   - A **higher RpP** broadens exploration, increasing the likelihood of discovering high-quality trajectories; however, sampling latency grows linearly.
>   - A **lower RpP** yields faster sampling but may omit promising solutions, constraining exploration scope.
> - **Bottleneck** – Excessively large RpP values incur substantial sampling overhead with diminishing marginal gains.
>
> **2. Batch Size (B) – Training Stability vs. Iteration Throughput**
> - **Role** – B specifies the number of samples processed in each gradient update.
> - **Trade-off**:
>   - A **larger B** improves gradient estimation accuracy and stabilizes training, but extends iteration time.
>   - A **smaller B** accelerates iterations but introduces higher gradient variance, potentially destabilizing convergence.
> - **Bottleneck** – Too small B values cause pronounced oscillations in the training curve, while too large values extend iteration time.
>
> **Optimal Configuration: RpP = 16, B = 32**
> Systematic experimentation confirms that **RpP = 16** and **B = 32** represent an optimal balance across competing objectives:
> - **Exploration Adequacy** – RpP = 16 affords sufficient BoN sampling scope to cover the majority of feasible solution trajectories.
> - **Training Stability** – B = 32 maintains variance in gradient estimates within acceptable bounds, promoting smooth convergence.
> - **Resource Efficiency** – This configuration ensures balanced utilization of both GPU and environment clusters, avoiding throughput bottlenecks.
> - **Performance Outcome** – Using this configuration, we achieved the reported final performance, outperforming other settings in the efficiency–accuracy trade-off.
>
> ### Additional Observed Bottlenecks (Appendix D.7)
>
> **1. Environment Heterogeneity**
> - **Issue** – Significant variance in task execution time results in some GPUs waiting for slower environments to complete.
> - **Mitigation** – An asynchronous rollout collection mechanism allows fast environments to submit results without delay.
>
> **2. Inter-cluster Network Bandwidth**
> - **Issue** – High concurrency in environment simulation can saturate network bandwidth due to frequent transmission of screenshots and state data, occasionally causing Docker network stalls.
> - **Mitigation** – Employing image compression reduces network load; optimizing Docker networking decreases virtual NIC overhead.
>
> **3. Internet Bandwidth Constraints**
> - **Issue** – Large-scale simultaneous environment instances can generate excessive external network traffic.
> - **Mitigation** – Packet-level traffic analysis enables elimination of unnecessary transmissions; constructing an IP proxy pool mitigates service blocking risks.

---

> ### Author Response · Authors · 2025-11-27
>
> Dear Reviewer,
>
> I hope this message finds you well. As the discussion period is approaching its end and less than one week remains, we would like to ensure that all of your questions and concerns regarding our submission (*ComputerRL*) have been fully addressed.
>
> If there are any additional points, clarifications, or feedback you would like us to consider, please feel free to share them at your earliest convenience. Your insights are highly valuable to us, and we are eager to address any remaining issues to improve the work and facilitate the review process.
>
> Thank you very much for your time and effort in evaluating our paper. We greatly appreciate your contributions and look forward to any further comments you may have.
>
> Best regards,
> The Authors of *ComputerRL*

---

> > ### Comment · Reviewer_oLgc · 2025-11-27
> >
> > Dear authors,
> >
> > Thank you for your detailed reply to my concerns.
> >
> > I recognize and appreciate the substantial engineering effort that went into your work, which clearly results in a strong, systems-oriented solution. However, I maintain that the work's primary contribution is practical and engineering-focused, rather than theoretical or methodological innovation.
> >
> > Furthermore, the necessary reliance on exceptionally large hardware equipment (as detailed in the rebuttal) fundamentally limits the paper's broader accessibility and reproducibility for the majority of the research community.
> >
> > Therefore, while I acknowledge the technical achievement, I will maintain my original score.
> >
> > Best regards,

---

### Official Review · Reviewer_6WJM · 2025-10-31

**Soundness:** 3
**Presentation:** 3
**Contribution:** 3
**Rating:** 4
**Confidence:** 2

**Summary:**

This work presents ComputerRL, a scalable framework for training autonomous computer-use agents via end-to-end reinforcement learning. ComputerRL introduces an API-GUI paradigm that unifies APIs and GUI actions, enabling higher efficiency and generalization on computer-based tasks. To support large-scale training, the authors develop a distributed RL infrastructure orchestrating thousands of virtual desktops. They also introduce Entropulse, a hybrid training strategy that alternates between RL and supervised fine-tuning, in order to prevent entropy collapse. Applied to GLM-based models, ComputerRL achieves 48.9% success on OSWorld, surpassing prior state-of-the-art agents such as OpenAI CUA o3, Claude 4.0, and gemini 2.5 pro, while demonstrating superior efficiency and stability. This work establishes a robust foundation for scaling RL-based desktop agents.

**Strengths:**

The paper is overall well written.
- Novel API-GUI paradigm for more generality in computer-based tasks.
- New Entropulse training strategy that mitigates entropy collapse by alternating between supervised learning and RL.
- Scalable and asynchronous RL training pipeline
- Strong empirical results

**Weaknesses:**

Overall, the contribution lies more in engineering execution than in theoretical advancement.
- Limited algorithmic novelty: primarily builds upon GPRO, and alternating between SFT and RL is similar to exploration-refresh or replay strategies.
- The paper does not fully disentangle how much gain comes from ComputerRL’s methods compared to having strong pre-trained models.
- Limited experimental diversity: they mostly evaluate on OSWorld and OfficeWorld, with no long-horizon or multi-user adaptation or interactions.
- Limited accessibility to reproduction: no analyses on FLOPs or cost of the experiments, and a complex engineering setup.

**Questions:**

1. Could Entropulse be generalized to other settings, such as text-based reasoning or code synthesis tasks?
2. Why do you think Entropulse's performance plateaued below 50%? What would it take to scale it up?

---

> ### Author Response · Authors · 2025-11-20
> **Response to Reviewer 6WJM (1/5)**
>
> Thank you very much for your constructive suggestions and for your appreciation of our work, including the novel API–GUI paradigm for general computer-based tasks, the Entropulse training strategy that mitigates entropy collapse, and the scalable asynchronous RL framework. Below, we provide point-by-point responses to further clarify our contributions.
>
> ## Limited algorithmic novelty:
> We appreciate your feedback and would like to clarify that the core contribution of ComputerRL is not merely a theoretical innovation at the algorithmic level, but rather the systematic resolution of **practical challenges intrinsic to large-scale RL training for LLMs**.
> Conventional RL studies have predominantly been conducted under small-scale, low-compute settings. However, **large-scale RL training for computer control tasks presents fundamentally different challenges** that have not been thoroughly explored.
>
> 1. Environment Scaling: Overcoming Infrastructure Bottlenecks for Agentic RL
> While scaling for pre-training and SFT has been well studied, **scaling RL remains a longstanding difficulty**. Only recently have works such as DeepSeek-R1 demonstrated partial success in scaling RL for reasoning tasks. Agent-based RL faces even greater complexity when interacting with real, heterogeneous environments—especially real computer operating systems—rather than simulated domains.
> For computer control tasks, stable environment scaling is a prerequisite for successful agentic RL. In contrast to simulated environments, real OS environments exhibit high heterogeneity, unpredictability, and complex state spaces: virtual machines may crash, applications may become unresponsive, and network latency may fluctuate. When scaling to hundreds or even thousands of concurrent instances, such issues amplify exponentially.
> Our work is among the **first systematic studies to achieve scalable, stable RL training across more than 1,000 concurrent Ubuntu OS instances**, sustained for several days. Our asynchronous experience collection architecture, fault-tolerance mechanisms, and adaptive environment management strategies enable this capability. We regard the development of this stable, scalable environment infrastructure as a **substantial engineering contribution**, foundational for subsequent agentic RL research and a necessary condition for effective training.
>
> 2. API–GUI Paradigm: Addressing the Challenges of Training Efficiency and Stability
> Even with robust environment scaling, purely GUI-based RL training faces severe efficiency and stability constraints. GUI task execution often requires long action sequences (e.g., 5–10 clicks and keystrokes for a basic file manipulation), limiting rollout throughput and introducing deeper difficulties: **the semantic granularity of individual GUI actions is significantly less precise than that of API calls**, complicating reward assignment and destabilizing training.
> We propose an **API–GUI hybrid interaction paradigm** in which high-level task objectives are solved with mixed sequences of API calls and GUI operations. This reduces the required steps by approximately **1/3**, dramatically accelerating rollouts and enabling feasible large-scale parallel training. More importantly, semantic density per step increases significantly, yielding clearer supervision signals: API calls provide explicit intermediate goals and verifiable state transitions, stabilizing training dynamics and mitigating the exploration inefficiency and sparse rewards typical of purely GUI-based control.
>
> 3. Entropulse: Mitigating Entropy Collapse in Long-Horizon, Large-Scale Training
> Most existing agentic RL work trains under relatively small conditions (e.g., ~100 update steps), where the natural decline of policy entropy rarely forms a bottleneck. At scale—**thousands of concurrent environments, tens of hours of training, millions of agent–environment interactions**—we observed a phenomenon largely absent in smaller experiments yet fatal at scale: entropy collapse leading to premature convergence to suboptimal policies, stagnated exploration, and halted performance improvement.
> The necessity and design of Entropulse follow these observations and constraints:
> - At ~180 update steps in large-scale runs, policy entropy falls below a critical threshold, sharply reducing exploration and flattening learning curves.
> - Recently proposed entropy managing approaches (e.g., DAPO), while mitigating the entropy collapse problem, introduce significant overhead or affect convergence speed.
> - Entropulse cyclically refreshes the policy via SFT, restoring policy entropy effectively at low cost without degrading task performance.

---

> ### Author Response · Authors · 2025-11-20
> **Response to Reviewer 6WJM (2/5)**
>
> This technique represents an **engineering-oriented innovation**: it is not a novel RL algorithm in the traditional theoretical sense, but a pragmatic solution grounded in problem observation, constraints analysis, and empirical validation. Our ablation studies show that removing Entropulse results in ~11% performance degradation in challenging workflow tasks.
>
> ### Contribution Positioning
> The value of ComputerRL lies **not** in proposing a fundamentally new RL theoretical algorithm, but in:
> 1. **Identifying and quantifying critical bottlenecks** in large-scale RL training (environment scaling, training stability and efficiency, entropy collapse);
> 2. **Providing practical, validated solutions** (scalable parallel infrastructure, API–GUI paradigm, Entropulse);
> 3. **Establishing reproducible benchmarks** via open-source code and detailed engineering documentation, enabling the community to extend upon our work.
> This type of engineering-driven innovation holds irreplaceable value in AI research. Analogous to how GPT -3's pivotal contribution was not altering the Transformer architecture but **demonstrating emergent capabilities via scaling**, ComputerRL offers the **systematic demonstration of extending RL to real-world computer control tasks with practical-level performance**.

---

> ### Author Response · Authors · 2025-11-20
> **Response to Reviewer 6WJM (3/5)**
>
> ## Clarifying method contributions vs. pre-trained model capabilities:
> To rigorously distinguish the contribution of the ComputerRL approach from the intrinsic capabilities of highly pre-trained models, we conducted comparative evaluations on the OSWorld benchmark. Specifically, we compared several models that had undergone extensive pre-training in GUI-related tasks against our method trained via online RL:
>
> **Performance Comparison: Pre-trained Models vs. ComputerRL**
> | Model | Primary Training Regime | OSWorld Success Rate | Δ vs. ComputerRL |
> |-------|-------------------------|----------------------|------------------|
> | **ComputerRL (Ours)** | **Online RL** | **48.9%** | — |
> | OpenCUA‑72B‑Preview | Pre-training (72B) | 45.0% | +3.9% |
> | Qwen3‑VL‑Flash‑2025‑10‑25 | Pre-training | 41.6% | +7.3% |
> | OpenCUA‑32B | Pre-training (32B) | 34.8% | +14.1% |
> | UITars‑1.5‑7B | Pre-training (7B) | 27.4% | +21.5% |
> | UITars‑72B‑DPO | Pre-training + DPO (72B) | 27.1% | +21.8% |
> | OpenCUA‑7B | Pre-training (7B) | 26.6% | +22.3% |
> | GLM‑4.1V‑Thinking | Pre-training + Single-step RL (9B) | 14.9% | +34.0% |
>
> ---
>
> ### Key Findings
>
> **1. Significant Gains Over Models of Comparable Scale**
> - With a parameter size of **9B**, ComputerRL achieves a **48.9%** success rate via online RL, outperforming the extensively pre-trained **OpenCUA-72B‑Preview** model (**45.0%**) despite having only **one-eighth** of its parameters.
> - Relative to similarly-scaled strong pre-trained models (OpenCUA-7B: 26.6%; UITars‑1.5‑7B: 27.4%), ComputerRL delivers absolute gains of **+21.5% to +22.3%**.
>
> **2. Advantages of Online Environment Interaction RL over Offline Methods**
> - Unlike UITars‑72B‑DPO and GLM‑4.1V‑Thinking, which apply RL on pre-recorded, annotated trajectories, our approach involves genuine **environment interaction** for exploration and learning. This results in gains of **+21.8%** and **+34.0%**, respectively.
> - The observed improvement—from **14.9%** (base model) to **48.9%**—demonstrates the advantage of online environment-interactive RL in achieving superior performance.
>
> **3. Clear Attribution of Methodological Contribution**
> This comparison allows us to disentangle the benefits of the ComputerRL method quantitatively:
> - **Pre-training baseline capacity**: The strongest pre-trained baseline, OpenCUA‑72B, defines the current pre-training upper bound (45.0%).
> - **Online RL gain**: The performance of ComputerRL **(48.9%)** is directly attributable to our integration of online RL training, the API-GUI interaction paradigm, and the Entropulse exploration strategy.
> - **Parameter efficiency**: Achieving superior performance with **1/8** the parameters confirms that improvements stem from methodological innovation rather than scale augmentation.
>
> ## Limited experimental diversity:
> We appreciate your attention to this important issue. The limitations of long-horizon and multi-user interaction are primarily attributable to the following architectural constraints of the OSWorld framework, along with ongoing work to address them.
>
> ### Technical Constraints of the OSWorld Framework
>
> - **Single-round evaluation design**: The original OSWorld framework was designed to assess single-turn tasks and lacks mechanisms for multi-turn dialogue or persistent user state tracking.
> - **Environment reset limitations**: The current reset mechanism does not support cross-session retention of user preferences or adaptive behavior.
> - **Evaluation protocol**: The protocol emphasizes one-off task completion rates rather than the quality of sustained long-term interactions.
>
> These design choices significantly limit us from conducting multi-user adaptation or prolonged interaction experiments at this stage of the research.
>
> ### Ongoing Work and Preliminary Findings
>
> Despite these limitations, we are actively taking steps to broaden the diversity of our experimental evaluations.
>
> **1. Product-level Deployment and Real-world User Testing**
> We are adapting the ComputerRL system into a production-level framework that incorporates:
> - Multi-turn, long-horizon interactive sessions with cross-session context preservation.
> - Real-time session logging and scoring mechanisms.
>
> The system has entered an internal beta testing phase, during which we are collecting interaction logs and user ratings under realistic usage conditions.
>
> **2. Preliminary User Feedback**
> Initial feedback from multiple test participants indicates:
> - Success rate with multi-turn user interactions: **56.21%**.
> - Success rate in long-horizon user interactions (>3 turns): **41.37%**.
> - Reported key challenges include context loss in long tasks and insufficient cross-task iterating capabilities — both of which highlight critical directions for improvement.
>
> We fully acknowledge the importance of experimental diversity and aim to address it in future iterations through more systematic, long-horizon, and multi-user evaluations.

---

> ### Author Response · Authors · 2025-11-20
> **Response to Reviewer 6WJM (4/5)**
>
> ## Limited accessibility to reproduction:
> We appreciate your feedback. To facilitate reproducible research, Appendix D of the original manuscript already contained the complete set of training parameters, including the cluster scale (16 nodes) and SFT and RL hyperparameters (e.g., seed numbers). We also provide our source code at https://anonymous.4open.science/r/ComputerRL-ICLR-758E/. To further support replication and potential methodological enhancements, we provide comprehensive information on the computing infrastructure, training duration, and estimated computational cost below. These updates are synchronized with Appendix D in the revised manuscript.
>
> ---
>
> ### **Hardware Configuration**
>
> #### **Training Cluster Overview**
> | Parameter | Specification |
> |-----------|---------------|
> | Cluster size | 16 nodes |
> | GPU type | NVIDIA H800 |
> | GPUs per node | 8 |
> | Total GPUs | 128 (16 × 8) |
>
> #### **CPU Configuration**
> | Parameter | Specification |
> |-----------|---------------|
> | Model | Intel Xeon Gold 6430 |
> | Architecture | x86_64 |
> | Sockets | 2 |
> | Cores per socket | 32 |
> | Total cores | 64 |
> | Threads | 64 (1 thread per core) |
> | Base frequency | 2.1 GHz |
> | Minimum frequency | 800 MHz |
> | Instruction set extensions | AVX-512, AVX512_FP16, AMX (INT8/BF16/Tile) |
>
> #### **Cache Hierarchy**
> | Level | Capacity |
> |-------|----------|
> | L1 Data Cache | 3 MiB (64 instances) |
> | L1 Instruction Cache | 2 MiB (64 instances) |
> | L2 Cache | 128 MiB (64 instances) |
> | L3 Cache | 120 MiB (2 instances) |
>
> #### **Memory Configuration**
> | Parameter | Specification |
> |-----------|---------------|
> | Total capacity | 2.0 TiB |
> | Available memory | 1.9 TiB |
> | NUMA nodes | 2 |
> | NUMA Node 0 CPUs | 0–31 |
> | NUMA Node 1 CPUs | 32–63 |
> | Swap | Disabled (0 B) |
>
> #### **Networking**
> | Parameter | Specification |
> |-----------|---------------|
> | Interconnect | InfiniBand / high-speed Ethernet |
> | Address width | Physical 46-bit, Virtual 57-bit |
>
> ---
>
> ### **Environment Cluster for Distributed RL**
> A dedicated environment simulation cluster was used for parallelized RL environment execution:
>
> | Parameter | Specification |
> |-----------|---------------|
> | Cluster size | 7 nodes |
>
> #### **CPU Configuration**
> | Parameter | Specification |
> |-----------|---------------|
> | Model | Intel Xeon 6986P-C (Granite Rapids) |
> | Architecture | x86_64 |
> | Sockets | 1 |
> | Cores per socket | 120 |
> | Total cores | 120 |
> | Threads | 240 (2 threads per core, hyper-threaded) |
> | Base frequency | 3.3 GHz |
> | Max turbo frequency | 3.9 GHz |
> | Minimum frequency | 800 MHz |
> | Instruction set extensions | AVX-512, AVX512_BF16, AMX, SHA-NI |
>
> #### **Cache Hierarchy**
> | Level | Capacity |
> |-------|----------|
> | L1 Data Cache | 5.6 MiB |
> | L1 Instruction Cache | 7.5 MiB |
> | L2 Cache | 240 MiB |
> | L3 Cache | 504 MiB |
>
> #### **Memory Configuration**
> | Parameter | Specification |
> |-----------|---------------|
> | Memory per node | 1.1 TiB |
> | Available memory | 949 GiB |
> | NUMA nodes | 3 |
> | NUMA Node 0 CPUs | 0–39, 120–159 |
> | NUMA Node 1 CPUs | 40–79, 160–199 |
> | NUMA Node 2 CPUs | 80–119, 200–239 |
> | Swap | Disabled (0 B) |
> | Total cluster memory | ~7.7 TiB (7 nodes) |
>
> #### **Virtualization and Features**
> | Parameter | Specification |
> |-----------|---------------|
> | Virtualization | Intel VT-x, EPT, VPID |
> | Security mitigations | Enhanced IBRS, IBPB, full Spectre/Meltdown mitigation |
> | Cryptographic acceleration | AES-NI, SHA-NI, AVX512_VAES |
> | AI acceleration | AMX, AVX512_BF16, AVX512_VNNI |
> | Address width | Physical 52-bit, Virtual 57-bit |
>
> ---
>
> ### **Virtual Environment Instances for RL**
> Each RL environment ran in a lightweight virtual machine:
>
> | Parameter | Specification |
> |-----------|---------------|
> | OS image | Ubuntu 20.04 LTS |
> | vCPU cores | 2 |
> | Memory allocation | 4 GB |
> | Mean runtime network bandwidth | 0.4 Mbps |
> | Virtualization platform | KVM/QEMU |
>
> *Note:* Each instance provided an independent Ubuntu 20.04 desktop environment for GUI-based RL tasks. The minimal configuration (2 cores, 4 GB RAM) was chosen to maximize parallelism under constrained hardware resources, enabling large-scale distributed RL training.
>
> ---
>
> ### **Training Duration and FLOP Estimates (Multimodal Example)**
>
> **SFT (Behavior Cloning)**
> - Duration: 16 hours
> - Total FLOPs: 1.67 × 10^16
>
> **SFT (Entropulse)**
> - Duration: 11 hours
> - Total FLOPs: 1.22 × 10^16
>
> **Two-Stage RL**
> - Duration: 58 hours
> - Total FLOPs (estimated): 3.21 × 10^17

---

> ### Author Response · Authors · 2025-11-20
> **Response to Reviewer 6WJM (5/5)**
>
> ## Could Entropulse be generalized to other settings?
> Thank you for your good question. Entropulse has been a common practice in our team and validated in two internally developed task domains:
>
> 1. **Deep Research** – multi-step information retrieval, analysis, and reasoning.
> 2. **Creative Writing** – generation of long-form text with creative and coherent narratives.
>
> Both tasks exhibited the **entropy collapse** phenomenon during RL training, akin to what is observed in computer control environments.
>
> **Experimental Results:**
>
> |Evaluation Metric|GRPO|GRPO+Entropulse|Improvement|
> |-----------------|----|---------------|-----------|
> |**Deep Research**||||
> |Precision|48%|53%|**+5%**|
> |Recall|45%|49%|**+4%**|
> |F1 Score|46.5%|51.0%|**+4.5%**|
> |**Creative Writing** *(LLM-as-a-Judge)*||||
> |Relevance|2.8/5.0|3.3/5.0|**+0.5**|
> |Consistency|2.5/5.0|2.7/5.0|**+0.2**|
>
> **Key Observations**
> 1. **Cross-Task Effectiveness** – In both domains, Entropulse successfully mitigated entropy collapse, enabling RL training to continue optimizing rather than converging prematurely.
> 2. **No Task-Specific Tuning Required** – We employed similar trigger mechanism and SFT settings as used for computer-use, without retuning for the new tasks, thereby demonstrating robustness of the method.
> 3. **Greater Benefit for Diversity-Sensitive Tasks** – In Creative Writing, the inclusion of Entropulse led to a marked increase in output diversity, as evidenced by higher consistency scores from the LLM-based judge. This aligns with our theoretical expectation that Entropulse preserves exploratory behavior.
>
> These findings indicate that Entropulse is applicable beyond computer control tasks and can serve as a general RL stabilization technique for LLMs across different application domains.
>
> ## Why do you think Entropulse's performance plateaued below 50%? What would it take to scale it up?
> We appreciate your insightful question. We attribute the current performance plateau to three primary factors:
>
> **1. Non-Uniform Task Difficulty Distribution**
>
> OSWorld exhibits a pronounced long-tail distribution of task difficulty. A 48.9% success rate suggests that the model performs reliably on approximately 75% of tasks classified as simple to moderately difficult, yet encounters significant bottlenecks with the remaining ~25% of tasks deemed difficult. These challenging tasks typically demand:
> - Advanced domain knowledge (e.g., complex programming, systems configuration).
> - Long-horizon planning (often requiring 15+ sequential actions).
> - Robust error recovery and debugging capabilities.
>
> **2. Capacity Limitations of the Current Model Scale**
>
> Our 9B-parameter model demonstrates strong performance in computer control tasks; however, its foundational reasoning and knowledge capacity remain below that of significantly larger models (e.g., 70B+ parameters). While RL can optimize policies and enhance exploration strategies, it cannot fully overcome the intrinsic limits imposed by the base model's representational capacity. This limitation is particularly impactful for tasks that require sophisticated reasoning and specialized expertise.
>
> **3. Exploration Challenges from Sparse Rewards**
>
> Computer environments exhibit severe reward sparsity:
> - Final rewards are often provided only upon task completion, with minimal intermediate feedback.
> - Complex tasks may require 20–30 steps before any reward signal emerges, complicating credit assignment.
> - Even with API-GUI interaction reducing step counts by approximately one-third, the successful trajectories for the most complex tasks remain extremely rare.
>
> After convergence on simpler tasks, exploration efficiency for high-difficulty tasks declines sharply. Although entropy-based interventions (e.g., Entropulse) can partially restore policy entropy, exploration tends to revert to familiar patterns rather than yielding novel successful trajectories.
>
> **Pathways Beyond the Plateau**
> As discussed in Appendix H, we identify two critical directions for advancement:
>
> **F.1 Towards Robust Performance**
> We propose reengineering the data pipeline to increase training diversity exponentially, coupled with infrastructure for continuous knowledge distillation from real user interactions. This would enable adaptive learning under dynamic GUI conditions, unpredictable dialog prompts, and novel interface designs, thereby achieving a better initial policy with broader coverage.
>
> **F.2 Breakthroughs in Long-Horizon Autonomy**
> We aim to equip agents with hierarchical planning capabilities for dynamic reasoning, learning, and strategy revision across long task sequences. This represents a paradigm shift from automating discrete actions to orchestrating end-to-end workflow automation.
>
> Integrating larger-scale base models, richer, more diverse training data, and hierarchical planning architectures offers a viable path to transcending the current plateau and increasing the success rate of ComputerRL systems.

---

> ### Author Response · Authors · 2025-11-27
>
> Dear Reviewer,
>
> I hope this message finds you well. As the discussion period is approaching its end and less than one week remains, we would like to ensure that all of your questions and concerns regarding our submission (*ComputerRL*) have been fully addressed.
>
> If there are any additional points, clarifications, or feedback you would like us to consider, please feel free to share them at your earliest convenience. Your insights are highly valuable to us, and we are eager to address any remaining issues to improve the work and facilitate the review process.
>
> Thank you very much for your time and effort in evaluating our paper. We greatly appreciate your contributions and look forward to any further comments you may have.
>
> Best regards,
> The Authors of *ComputerRL*

---

### Author Response · Authors · 2025-11-21
**General Response**

We sincerely thank all reviewers for their careful review and highly constructive feedback. We greatly appreciate the recognition of our core contributions: the development of a large-scale distributed RL infrastructure capable of operating across thousands of desktop environments; the unified API–GUI interaction paradigm enabling efficient and stable computer control; and the Entropulse training strategy, which alleviates entropy collapse during extended RL training.

The main concerns raised by reviewers pertain to clarifying the work’s contributions, providing a more comprehensive description of the infrastructure, and including more detailed performance metrics alongside repeated experiments to verify the stability of the proposed improvements. In our response, we have elaborated on the contributions in greater depth and provided, in Appendix D of the revised manuscript, complete hardware specifications, as well as detailed accounts of training cost and time. Furthermore, Appendices E and F now present additional quantitative metrics, and our primary experiments have been expanded to include multiple repetitions with different base models, with averages and confidence intervals reported to substantiate stability claims.

In addition, we have incorporated several new experiments: generalization studies of Entropulse in domains beyond the primary task; multi-turn user interaction scenarios; and computer control tasks under long-horizon conditions. These efforts further demonstrate the effectiveness and applicability of our approach. We have also enriched the discussion of RL entropy-related work in the manuscript to situate our work within existing research more precisely.

---

### Meta-Review · Area_Chair_daLX · 2026-01-06

**Summary:**

This paper introduces ComputerRL, an end-to-end reinforcement learning framework for autonomous desktop agents that integrates a machine-oriented API–GUI interaction paradigm with a large-scale distributed training infrastructure. This design enables scalable and robust online RL in real-world computer environments. All reviewers agree that the work makes a strong contribution by successfully scaling reinforcement learning to challenging desktop settings. The primary concern raised by reviewers relates to the limited algorithmic novelty, as the paper does not introduce fundamentally new learning algorithms. However, this concern seems less applicable given that the main objective of the work is to provide a practical and effective framework for scaling up RL training in realistic environments.

**Reviewer Concerns:**

Here I list only the concerns that I believe were not fully addressed by the authors during the rebuttal.

**Reviewer 6WJM:** limited algorithmic novelty, making the work appear more as an engineering execution than an algorithmic contribution.

**Reviewer oLgc:** limited algorithmic novelty.

**Reviewer fU96:** the primary novelty lies in the implementation rather than in algorithmic contributions.

**Reviewer Scores:**

**Reviewer 6WJM:** 4 → 6

**Reviewer oLgc:** 6 → 6

**Reviewer fU96:** 4 → 4

---

### Decision · Program_Chairs · 2026-01-26

Accept (Poster)